# FERMI: Fair Empirical Risk Minimization Via Exponential Rényi Mutual Information

## Abstract

Despite the success of large-scale empirical risk minimization (ERM) at achieving high accuracy across a variety of machine learning tasks, fair ERM is hindered by the incompatibility of fairness constraints with stochastic optimization. In this paper, we propose the fair empirical risk minimization via exponential Rényi mutual information (FERMI) framework. FERMI is built on a stochastic estimator for exponential Rényi mutual information (ERMI), an information divergence measuring the degree of the dependence of predictions on sensitive attributes. Theoretically, we show that ERMI upper bounds existing popular fairness violation metrics, thus controlling ERMI provides guarantees on other commonly used violations, such as $L_\infty$. We derive an unbiased estimator for ERMI, which we use to derive the FERMI algorithm. We prove that FERMI converges for demographic parity, equalized odds, and equal opportunity notions of fairness in stochastic optimization. Empirically, we show that FERMI is amenable to large-scale problems with multiple (non-binary) sensitive attributes and non-binary targets. Extensive experiments show that FERMI achieves the most favorable tradeoffs between fairness violation and test accuracy across all tested setups compared with state-of-the-art baselines for demographic parity, equalized odds, equal opportunity. These benefits are especially significant for non-binary classification with large sensitive sets and small batch sizes, showcasing the effectiveness of the FERMI objective and the developed stochastic algorithm for solving it.

## 1 Introduction

Ensuring that decisions made using machine learning algorithms are fair to different subgroups is of utmost importance. Without any mitigation strategy, machine learning algorithms may result in discrimination against certain subgroups based on sensitive attributes, such as gender or race, even if such discrimination is absent in the training data (Datta et al., 2015; Sweeney, 2013; Bolukbasi et al., 2016; Angwin et al., 2016; Calmon et al., 2017b; Feldman et al., 2015; Hardt et al., 2016; Fish et al., 2016; Woodworth et al., 2017; Zafar et al., 2017; Bechavod & Ligett, 2017; Kearns et al., 2018). Algorithmic fairness literature aims to remedy such discrimination issues.

A machine learning algorithm satisfies the *demographic parity* fairness notion, if the predicted target is independent of the sensitive attributes (Dwork et al., 2012). Promoting demographic parity can lead to poor performance, especially if the true outcome is not independent of the sensitive attributes. To remedy this, Hardt et al. (2016) proposed *equalized odds* to ensure that the predicted target is conditionally independent of the sensitive attributes given the true label. A further relaxed version of this notion is *equal opportunity* which is satisfied if predicted target is conditionally independent of sensitive attributes given that the true label is in an advantaged class (Hardt et al., 2016). The inherent assumption in such conditional notions is that the true labels are fair. These notions suffer from a potential amplification of the inherent discrimination that may exist in the training data. Tackling such bias is beyond the scope of this work; cf. Kilbertus et al. (2020) and Bechavod et al. (2019).

Submitted to 35th Conference on Neural Information Processing Systems (NeurIPS 2021). Do not distribute.

| Reference | NB target | NB attrib. | NB code | Fairness notion dp | eod | eop | Beyond logistic | Stoch. alg. (unbiased**) | Converg. (stoch.) |
|---|---|---|---|---|---|---|---|---|---|
| **FERMI (this work)** | ✓ | ✓ | ✓ | ✓ | ✓ | ✓ | ✓ | ✓ (✓) | ✓ (✓) |
| (Cho et al., 2020b) | ✓ | ✓ | ✓ | ✓ | ✓ | ✗ | ✓ | ✓ (✗) | ✗ |
| (Cho et al., 2020a) | ✓ | ✓ | ✗ | ✓ | ✓ | ✗ | ✓ | ✓ (✓) | ✗ |
| (Baharlouei et al., 2020) | ✓ | ✓ | ✓ | ✓ | ✓ | ✓ | ✓ | ✗ | ✓ (✗) |
| (Rezaei et al., 2020) | ✗ | ✗ | ✗ | ✓ | ✓ | ✗ | ✗ | ✗ | ✗ |
| (Jiang et al., 2020)* | ✗ | ✗ | ✗ | ✓ | ✗ | ✗ | ✗ | ✗ | ✗ |
| (Mary et al., 2019) | ✓ | ✓ | ✓ | ✓ | ✓ | ✗ | ✓ | ✓ (✗) | ✗ |
| (Donini et al., 2018) | ✗ | ✓ | ✗ | ✗ | ✓ | ✗ | ✓ | ✗ | ✗ |
| (Zhang et al., 2018) | ✓ | ✓ | ✗ | ✓ | ✓ | ✗ | ✓ | ✓ (✗) | ✗ |

Table 1: Comparison of state-of-the-art in-processing methods. **NB = non-binary,** dp = demographic parity, eod = equalized odds, eop = equal opportunity. While satisfying eod guarantees satisfying eop, an eod algorithm does not necessarily achieve a favorable tradeoff between performance and fairness violation in eop; we only credit those works that provide/implement algorithms for a given fairness notion. FERMI is the only method compatible with stochastic optimization and guaranteed convergence. The only existing baselines for non-binary classification with non-binary sensitive attributes are (Mary et al., 2019; Baharlouei et al., 2020; Cho et al., 2020b) (NB code). *We refer to the in-processing method of (Jiang et al., 2020), not their post-processing method. **We use the term "unbiased" to refer to unbiased estimation in statistical sense; it is not to be confused with bias in the fairness sense, for which we use the term discrimination.

**Measuring fairness violation.** In practice, the learner only has access to finite samples and cannot verify demographic parity, equalized odds, or equal opportunity. This has led the machine learning community to define several fairness violation metrics that quantify the degree of (conditional) independence between random variables, e.g., $L_\infty$ distance (Dwork et al., 2012; Hardt et al., 2016), mutual information (Kamishima et al., 2011; Rezaei et al., 2020; Steinberg et al., 2020; Zhang et al., 2018; Cho et al., 2020a), Pearson correlation (Zafar et al., 2017), false positive/negative rates (Bechavod & Ligett, 2017), Hilbert Schmidt independence criterion (HSIC) (Pérez-Suay et al., 2017), Rényi correlation (Mary et al., 2019; Baharlouei et al., 2020; Grari et al., 2019, 2020), and exponential Rényi mutual information (ERMI) (Mary et al., 2019). In this paper, we focus on three variants of ERMI specialized to demographic parity, equalized odds, and equal opportunity. We prove that ERMI provides an upper bound on the rest of the above existing notions of fairness violation. Consequently, a model trained to reduce ERMI will also provide guarantees on these other fairness violations. We also develop a stochastic estimator for ERMI that is compatible with large-scale stochastic optimization, and use it as a regularizer in within ERM, and call it FERMI. We theoretically show that FERMI is convergent, and empirically demonstrate that it outperforms all other state-of-the-art baselines, including (Mary et al., 2019) which solves the same objective as FERMI.

**Related work & contributions.** Fairness-promoting machine learning algorithms can be categorized in three main classes: *pre-processing*, *post-processing*, and *in-processing* methods. Pre-processing algorithms (Feldman et al., 2015; Zemel et al., 2013; Calmon et al., 2017b) transform the biased data features to a new space in which the labels and sensitive attributes are statistically independent. This transform is oblivious to the training procedure. Post-processing approaches (Hardt et al., 2016; Pleiss et al., 2017) mitigate the discrimination of the classifier by altering the the final decision. In-processing approaches focus on the training procedure and impose the notions of fairness as constraints or regularization terms in the training procedure. Several regularization-based methods are proposed in the literature to promote fairness in decision-trees (Kamiran et al., 2010; Raff et al., 2018; Aghaei et al., 2019), support vector machines (Donini et al., 2018), neural networks (Grari et al., 2020; Cho et al., 2020b), or (logistic) regression models (Zafar et al., 2017; Berk et al., 2017; Taskesen et al., 2020; Chzhen & Schreuder, 2020; Baharlouei et al., 2020; Jiang et al., 2020; Grari et al., 2019). While in-processing approaches generally give rise to better tradeoffs between fairness violation and performance, existing approaches are mostly incompatible with large-scale stochastic optimization. This paper addresses this problem. See below for a summary of our contributions and Table 1 for a summary of the main differences between FERMI and existing in-processing methods.

1. We analyze a notion of fairness violation called ERMI. We show that ERMI is a stronger notion of fairness violation than all existing notions. Therefore, a model that ensures small ERMI violation is guaranteed to have small fairness violation with respect to all other notions as well.

2. We formulate an empirical objective, called FERMI objective, for using ERMI as a regularizer with empirical risk minimization. We propose a solver for FERMI, which is the first stochastic in-processing fairness algorithm with guaranteed convergence. The existing stochastic fairness

algorithms by Zhang et al. (2018); Mary et al. (2019); Cho et al. (2020a,b) are not guaranteed to converge.

3. We demonstrate through extensive numerical experiments that FERMI achieves superior fairness-accuracy tradeoff curves against all comparable baselines, even when fairness violation is measured in terms of commonly used $L_\infty$ (for demographic parity, equalized odds, and equal opportunity). In particular, the performance gap is very large when minibatch size is small (as is practically necessary for large-scale problems), and the number of sensitive attributes is large.

## 2 Fairness notions: demographic parity, equalized odds, equal opportunity

In this section, we state a notion of fairness that generalizes demographic parity, equalized odds, and equal opportunity fairness definitions (the three notions considered in this paper). This will be convenient for presenting our theoretical results. Consider a learner who trains a model to make a prediction, $\widehat{Y}$, e.g., whether or not to extend a loan, supported on $\mathcal{Y}$ which can be discrete or continuous. The prediction is made using a set of features, $\mathbf{X}$, e.g., financial history features. We assume that there is a set of discrete sensitive attributes, $S$, e.g., race and sex, supported on $\mathcal{S}$, associated with each sample. Further, let $\mathcal{A} \subseteq \mathcal{Y}$ denote an advantaged outcome class, e.g., the outcome where a loan is extended.

**Definition 1** (($Z, \mathcal{Z}$)-fairness). *Given a random variable $Z$, let $\mathcal{Z}$ be a subset of values that $Z$ can take. We say that a learning machine satisfies $(Z, \mathcal{Z})$-fairness if for every $z \in \mathcal{Z}$, $\widehat{Y}$ is conditionally independent of $S$ given $Z = z$, i.e. $\forall \widehat{y} \in \mathcal{Y}, s \in \mathcal{S}, z \in \mathcal{Z}, p_{\widehat{Y},S|Z}(\widehat{y},s|z) = p_{\widehat{Y}|Z}(\widehat{y}|z)p_{S|Z}(s|z)$.*

($Z, \mathcal{Z}$)-fairness includes the popular demographic parity, equalized odds, and equal opportunity notions of fairness as special cases:

1. ($Z, \mathcal{Z}$)-fairness recovers demographic parity (Dwork et al., 2012) if $Z = 0$ and $\mathcal{Z} = \{0\}$. In this case, conditioning on $Z$ has no effect, and hence $(0, \{0\})$ fairness is equivalent to the independence between $\widehat{Y}$ and $S$ (see Definition 6, Appendix A).

2. ($Z, \mathcal{Z}$)-fairness recovers equalized odds (Hardt et al., 2016) if $Z = Y$ and $\mathcal{Z} = \mathcal{Y}$. In this case, $Z \in \mathcal{Z}$ is trivially satisfied. Hence, conditioning on $Z$ is equivalent to conditioning on $Y$, which recovers the equalized odds notion of fairness, i.e., conditional independence of $\widehat{Y}$ and $S$ given $Y$ (see Definition 7, Appendix A).

3. ($Z, \mathcal{Z}$)-fairness recovers equal opportunity (Hardt et al., 2016) if $Z = Y$ and $\mathcal{Z} = \mathcal{A}$. This is also similar to the previous case with $\mathcal{Y}$ replaced with $\mathcal{A}$ (see Definition 8, Appendix A).

Note that verifying $(Z, \mathcal{Z})$-fairness requires having access to the joint distribution of random variables $(Z, \widehat{Y}, S)$. This joint distribution is unavailable to the learner in the context of machine learning, and hence the learner would resort to empirical estimation of the amount of violation of independence, measured through some divergence. See (Williamson & Menon, 2019) for a related discussion.

## 3 Measuring fairness violation using exponential Rényi mutual information

Most existing fairness violations can be viewed as a (conditional) $f$-divergence between the joint distribution of sensitive attributes and predicted targets, $p_{\widehat{Y},S|Z}$, and the Kronecker proudct of the marginals, $p_{\widehat{Y}|Z} \otimes p_{S|Z}$. In this section, we focus on ERMI and show that several existing fairness violations are upper bounded by ERMI. For brevity, we present all definitions and results $(Z, \mathcal{Z})$.

**Definition 2** (ERMI – exponential Rényi mutual information). *We define the exponential Rényi mutual information between $\widehat{Y}$ and $S$ given $Z \in \mathcal{Z}$ as*

$$D_R(\widehat{Y}; S|Z \in \mathcal{Z}) := \mathbb{E}_{Z,\widehat{Y},S} \left\{ \frac{p_{\widehat{Y},S|Z}(\widehat{Y},S|Z)}{p_{\widehat{Y}|Z}(\widehat{Y}|Z)p_{S|Z}(S|Z)} \middle| Z \in \mathcal{Z} \right\} - 1. \tag{ERMI}$$

In Appendix B, we unravel the definition for the special cases of interest corresponding to demographic parity, equalied odds, and equal opportunity. We also discuss that ERMI is the $\chi^2$-divergence (which is an $f$-divergence) between the joint distribution, $p_{\widehat{Y},S|Z}$, and the Kronecker product of marginals, $p_{\widehat{Y}|Z} \otimes p_{S|Z}$ (Calmon et al., 2017a). In particular, ERMI is non-negative, and zero if and only if $(Z, \mathcal{Z})$-fairness is satisfied. In the context of algorithmic fairness, ERMI was first used by Mary et al. (2019) as a regularizer. We will provide a new stochastic solver/estimator for ERMI, which theoretically converges and empirically outperforms the one by Mary et al. (2019).

125 **Definition 3** (Rényi mutual information ([Rényi, 1961])). *Let the Rényi mutual information of order*
126 $\alpha > 1$ *between random variables $\widehat{Y}$ and $S$ given $Z \in \mathcal{Z}$ be defined as:*

$$I_\alpha(\widehat{Y}; S | Z \in \mathcal{Z}) := \frac{1}{\alpha - 1} \log \left( \mathbb{E}_{Z, \widehat{Y}, S} \left\{ \left( \frac{p_{\widehat{Y}, S | Z}(\widehat{Y}, S | Z)}{p_{\widehat{Y} | Z}(\widehat{Y} | Z) p_{S | Z}(S | Z)} \right)^{\alpha - 1} \middle| Z \in \mathcal{Z} \right\} \right), \qquad \text{(RMI)}$$

127 *which generalizes Shannon mutual information*

$$I_1(\widehat{Y}; S | Z \in \mathcal{Z}) := \mathbb{E}_{Z, \widehat{Y}, S} \left\{ \log \left( \frac{p_{\widehat{Y}, S | Z}(\widehat{Y}, S | Z)}{p_{\widehat{Y} | Z}(\widehat{Y} | Z) p_{S | Z}(S | Z)} \right) \middle| Z \in \mathcal{Z} \right\}, \qquad \text{(MI)}$$

128 *and recovers it as $\lim_{\alpha \to 1^+} I_\alpha(\widehat{Y}; S | Z \in \mathcal{Z}) = I_1(\widehat{Y}; S | Z \in \mathcal{Z})$.*

129 Note that $I_\alpha(\widehat{Y}; S | Z \in \mathcal{Z}) \geq 0$ with equality if and only if $(Z, \mathcal{Z})$-fairness is satisfied.
130 **Theorem 1** (ERMI is stronger than Shannon mutual information). *We have*

$$0 \leq I_1(\widehat{Y}; S | Z \in \mathcal{Z}) \leq I_2(\widehat{Y}; S | Z \in \mathcal{Z}) \leq e^{I_2(\widehat{Y}; S | Z \in \mathcal{Z})} - 1 = D_R(\widehat{Y}; S | Z \in \mathcal{Z}). \qquad (1)$$

131
132 All proofs are relegated to the appendix. Theorem 1 establishes that ERMI is a stronger measure of
133 fairness violation in the sense that driving it to zero would also bound the Shannon mutual information,
134 which is used for promoting fairness in recent literature ([Cho et al., 2020a]). It also shows that ERMI
135 is exponentially related to the Rényi mutual information of order 2.
136 **Definition 4** (Rényi correlation ([Hirschfeld, 1935; Gebelein, 1941; Rényi, 1959])). *Let $\mathcal{F}$ and $\mathcal{G}$*
137 *be the set of measurable functions such that for random variables $\widehat{Y}$ and $S$, $\mathbb{E}_{\widehat{Y}}\{f(\widehat{Y}; z)\} =$*
138 $\mathbb{E}_S\{g(S; z)\} = 0, \mathbb{E}_{\widehat{Y}}\{f(\widehat{Y}; z)^2\} = \mathbb{E}_S\{g(S; z)^2\} = 1,$ *for all $z \in \mathcal{Z}$. Rényi correlation is:*

$$\rho_R(\widehat{Y}, S | Z \in \mathcal{Z}) := \sup_{f, g \in \mathcal{F} \times \mathcal{G}} \mathbb{E}_{Z, \widehat{Y}, S} \left\{ f(\widehat{Y}; Z) g(S; Z) \middle| Z \in \mathcal{Z} \right\}. \qquad \text{(RC)}$$

139
140 Rényi correlation generalizes Pearson correlation,

$$\rho(\widehat{Y}, S | Z \in \mathcal{Z}) := \mathbb{E}_Z \left\{ \frac{\mathbb{E}_{\widehat{Y}, S}\{\widehat{Y} S | Z\}}{\sqrt{\mathbb{E}_{\widehat{Y}}\{\widehat{Y}^2 | Z\} \mathbb{E}_S\{S^2 | Z\}}} \middle| Z \in \mathcal{Z} \right\}, \qquad \text{(PC)}$$

141
142 to capture nonlinear dependencies between the random variables by finding functions of random
143 variables that maximize the Pearson correlation coefficient between the random variables. In fact,
144 it is true that $\rho_R(\widehat{Y}, S | Z \in \mathcal{Z}) \geq 0$ with equality if and only if $(Z, \mathcal{Z})$-fairness is satisfied. Rényi
145 correlation has gained popularity as a measure of fairness violation ([Mary et al., 2019; Baharlouei
146 et al., 2020; Grari et al., 2020]). Rényi correlation is also upper bounded by ERMI. The following
147 result has already been shown by [Mary et al. (2019)] and we present it for completeness.
148 **Theorem 2** (ERMI is stronger than Rényi correlation). *We have*

$$0 \leq |\rho(\widehat{Y}, S | Z \in \mathcal{Z})| \leq \rho_R(\widehat{Y}, S | Z \in \mathcal{Z}) \leq D_R(\widehat{Y}; S | Z \in \mathcal{Z}), \qquad (2)$$

149 *and if $|\mathcal{S}| = 2$, $D_R(\widehat{Y}; S | Z \in \mathcal{Z}) = \rho_R(\widehat{Y}, S | Z \in \mathcal{Z})$.*
150 **Definition 5** ($L_q$ fairness violation). *We define the $L_q$ fairness violation for $q \geq 1$ by:*

$$L_q(\widehat{Y}, S | Z \in \mathcal{Z}) := \mathbb{E}_Z \left\{ \left( \int_{\widehat{y} \in \mathcal{Y}_0} \sum_{s \in \mathcal{S}_0} \left| p_{\widehat{Y}, S | Z}(\widehat{y}, s | Z) - p_{\widehat{Y} | Z}(\widehat{y} | Z) p_{S | Z}(s | Z) \right|^q dy \right)^{\frac{1}{q}} \middle| Z \in \mathcal{Z} \right\}.$$
$$\text{(Lq)}$$

151 Note that $L_q(\widehat{Y}, S | Z \in \mathcal{Z}) = 0$ if and only if $(Z, \mathcal{Z})$-fairness is satisfied. In particular, $L_\infty$ fairness
152 violation recovers demographic parity violation ([Kearns et al., 2018], Definition 2.1) if we let $\mathcal{Z} = \{0\}$
153 and $Z = 0$. It also recovers equal opportunity violation ([Hardt et al., 2016]) if $\mathcal{Z} = \mathcal{A}$ and $Z = Y$.

154 **Theorem 3** (ERMI is stronger than $L_\infty$ fairness violation). *Let $\widehat{Y}$ be a discrete or continuous random*
155 *variable, and $S$ be a discrete random variable supported on a finite set. Then for any $q \geq 1$,*

$$0 \leq L_q(\widehat{Y}, S | Z \in \mathcal{Z}) \leq \sqrt{D_R(\widehat{Y}, S | Z \in \mathcal{Z})}. \qquad (3)$$

156

The above theorem says that if a method controls ERMI value for imposing fairness, then $L_\infty$ violation is controlled. In particular, the variant of ERMI that is specialized to demographic parity also controls $L_\infty$ demographic parity violation (Kearns et al., 2018). The variant of ERMI that is specialized to equal opportunity also controls the $L_\infty$ equal opportunity violation (Hardt et al., 2016). While our algorithm uses ERMI as a regularizer, in our experiments, we measure fairness violation through the more commonly used $L_\infty$ violation. Despite this, we show that our approach leads to better tradeoff curves between fairness violation and performance.

**Remark.** The bounds in Theorems 1-3 are not tight in general, but this is not of practical concern. They show that bounding ERMI is sufficient because any model that achieves small ERMI is guaranteed to satisfy any other fairness violation. This makes ERMI an effective regularizer for promoting fairness. In fact, in Sec. 5, we see that the proposed algorithm, FERMI, achieves the best tradeoffs between fairness violation and performance across state-of-the-art baselines.

# 4 FERMI: fair empirical risk minimization through ERMI regularization

Our goal is to train a model that balances fairness and accuracy objectives. To this end, we introduce fair risk minimization through exponential Rényi mutual information framework defined below:[1]

$$\min_{\boldsymbol{\theta}} \left\{ \text{FRMI}(\boldsymbol{\theta}) := \mathbb{E}_{\mathbf{X},Y,S} \left\{ \ell\big(\mathbf{X}, Y; \boldsymbol{\theta}\big) \right\} + \lambda D_R\big(\widehat{Y}(\mathbf{X}; \boldsymbol{\theta}); S\big) \right\}, \qquad \text{(FRMI obj.)}$$

where $\ell$ denotes the loss function, such as $L_2$ loss or cross entropy loss; $\lambda > 0$ is a scalar balancing the accuracy versus fairness objectives; $D_R\big(\widehat{Y}(\mathbf{X}; \boldsymbol{\theta}); S\big)$ is the notion of ERMI given in Eq. (ERMI) particularized to demographic parity (see Eq. (5)); and $\widehat{Y}(\mathbf{X}; \boldsymbol{\theta})$ is the output of the learned model (e.g., the output of a classification or a regression task, or the cluster number in a clustering task). While $\widehat{Y}(\mathbf{X}; \boldsymbol{\theta})$ inherently depends on $\mathbf{X}$ and $\boldsymbol{\theta}$, in the rest of this paper, we sometimes leave the dependence of $\widehat{Y}$ on $\mathbf{X}$ and/or $\boldsymbol{\theta}$ implicit for brevity of notation. Notice that we have also left the dependence of the loss on the predicted outcome $\widehat{Y}$ implicit.

In practice, the true joint distribution of $(\mathbf{X}, S, Y, \widehat{Y})$ is unknown and we only have $N$ samples at our disposal, making it impossible to solve FRMI. Let $\{\mathbf{x}_i, s_i, y_i, \widehat{y}_i(\mathbf{x}_i; \boldsymbol{\theta})\}_{i \in [N]}$ denote the features, sensitive attributes, targets, and the predictions of the model parameterized by $\boldsymbol{\theta}$ for these samples. Mary et al. (2019) considered the same objective Eq. (FRMI obj.), and tried to empirically solve it through a kernel approximation. We propose a completely different approach to solving this problem: fair empirical risk minimization via exponential Rényi mutual information (FERMI). FERMI results in a provably convergent algorithm, and empirically outperforms the algorithm by Mary et al. (2019). It is straightforward to derive an unbiased estimate for $\mathbb{E}_{\mathbf{X},Y,S} \left\{ \ell\big(\mathbf{X}, Y; \boldsymbol{\theta}\big) \right\}$ through the empirical risk, e.g., $\frac{1}{|B|} \sum_{i \in B} \ell\big(\mathbf{x}_i, y_i; \boldsymbol{\theta}\big)$ where $B \subseteq [N]$ is a random minibatch of data points. However, estimating $D_R(\widehat{Y}, S)$ in the objective function in Eq. (FRMI obj.) is more difficult. In what follows, we present our approach to deriving an *unbiased stochastic estimator* of $D_R(\widehat{Y}, S)$ given a random batch of data points $B$. The following theorem is the key tool we use to obtain an unbiased estimator:

**Theorem 4.** *For discrete random variables $\widehat{Y} = \widehat{Y}(\mathbf{X}; \boldsymbol{\theta})$ and $S$ where $\widehat{Y} \in [m], S \in [k]$, we have*

$$D_R(\widehat{Y}; S) = \max_{W \in \mathbb{R}^{k \times m}} \left\{ -\text{Tr}(W P_{\widehat{y}} W^T) + 2 \text{Tr}(W P_{\widehat{y},s} P_s^{-1/2}) - 1 \right\}, \qquad (4)$$

*where $P_{\widehat{y}} = \text{diag}(p_{\widehat{Y}}(1), \ldots, p_{\widehat{Y}}(m))$, $P_s = \text{diag}(p_S(1), \ldots, p_S(k))$, and*

$$P_{\widehat{y},s} = \begin{pmatrix} p_{\widehat{Y},S}(1,1) & \cdots & p_{\widehat{Y},S}(1,k) \\ \vdots & \ddots & \vdots \\ p_{\widehat{Y},S}(m,1) & \cdots & p_{\widehat{Y},S}(m,k) \end{pmatrix}.$$

Let $\widehat{\mathbf{Y}}, \widehat{\mathbf{y}}_i \in \{0,1\}^m$ and $\mathbf{S}, \mathbf{s}_i \in \{0,1\}^k$ be the one-hot encodings of $\widehat{Y}, \widehat{y}_i$ and $S, s_i$, respectively. Then, the above theorem implies that we can compute an unbiased estimate of Eq. (FRMI obj.):

---

[1]In this section, we present all results in the context of $Z = 0$ and $\mathcal{Z} = \{0\}$ (demographic parity), leaving off all conditional expectations for clarity of presentation. The results are readily generalized for general $(Z, \mathcal{Z})$ by using $D_R(\widehat{Y}, S | Z \in \mathcal{Z})$ in Eq. (FRMI obj.); we have used the resulting algorithms for empirical experiments.

**Lemma 1** (Unbiased estimator of ERMI). *Let* $(\mathbf{X}, S, Y, \widehat{Y}(\mathbf{X}; \boldsymbol{\theta}))$ *be a random draw from* $P_{\mathbf{X}, S, Y, \widehat{Y}}$. *Further, let*

$$\psi(\mathbf{X}, S, Y, \widehat{Y}; \boldsymbol{\theta}, W) := - \operatorname{Tr}(W \widehat{\mathbf{Y}}(\mathbf{X}; \boldsymbol{\theta}) \widehat{\mathbf{Y}}^T(\mathbf{X}; \boldsymbol{\theta}) W^T) + 2 \operatorname{Tr}(W \widehat{\mathbf{Y}}(\mathbf{X}; \boldsymbol{\theta}) \mathbf{S}^T P_s^{-1/2}) - 1.$$

*Then,* $\max_{W \in \mathbb{R}^{k \times m}} \psi(\mathbf{X}, S, Y, \widehat{Y}; \boldsymbol{\theta}, W)$ *is an unbiased estimator of ERMI in Eq.* (FRMI obj.)*, i.e.,*

$$\mathbb{E}_{\mathbf{X}, S, Y} \left\{ \max_{W \in \mathbb{R}^{k \times m}} \psi(\mathbf{X}, S, Y, \widehat{Y}; \boldsymbol{\theta}, W) \right\} = D_R(\widehat{Y}(\mathbf{X}; \boldsymbol{\theta}); S).$$

The stochastic estimator, $\psi(\mathbf{X}, S, Y, \widehat{Y}; \boldsymbol{\theta}, W)$, in Lemma 1 requires the knowledge of $P_s$, and computation of $P_s^{-1/2}$. This can be estimated with high fidelity (for small to moderate sensitive set) through a single initial pass over the entire dataset in practice. Hence, we consider it to be known. Now, we are equipped to state the empirical objective function that we solve in this paper:

$$\min_{\boldsymbol{\theta}} \max_{W \in \mathbb{R}^{k \times m}} \left\{ \text{FERMI}(\boldsymbol{\theta}, W) := \frac{1}{N} \sum_{i \in [N]} [\ell(\mathbf{x}_i, y_i; \boldsymbol{\theta}) + \lambda \psi_i(\boldsymbol{\theta}, W)] \right\}, \qquad \text{(FERMI obj.)}$$

where

$$\psi_i(\boldsymbol{\theta}, W) := - \operatorname{Tr}(W \widehat{\mathbf{y}}_i(\mathbf{x}_i; \boldsymbol{\theta}) \widehat{\mathbf{y}}_i^T(\mathbf{x}_i; \boldsymbol{\theta}) W^T) + 2 \operatorname{Tr}(W \widehat{\mathbf{y}}_i(\mathbf{x}_i; \boldsymbol{\theta}) \mathbf{s}_i^T P_s^{-1/2}) - 1.$$

In particular, Lemma 1 says that, for any $N$, Eq. (FERMI obj.) (and its gradients) is an *unbiased* and *consistent* estimator of the Eq. (FRMI obj.) objective function (and its gradients) by an empirical average over the minibatch. This is in contrast to the density estimation methods used by Mary et al. (2019) and Baharlouei et al. (2020), which are biased but consistent. We will see in the experiments that the unbiased estimator empirically offers large performance improvements.

This observations leads us to deriving a stochastic algorithm, presented in Algorithm 1, which is guaranteed to converge for any batch size $1 \leq |B| \leq N$ since the stochastic gradients are unbiased.

---

**Algorithm 1** (FERMI Algorithm). Two-Time Scale SGDA for solving FERMI objective

---

1: **Input**: $\boldsymbol{\theta}^0 \in \mathbb{R}^{d_\theta}$, $W^0 \in \mathcal{W} \subset \mathbb{R}^{k \times m}$, step-sizes $(\eta_\theta, \eta_w)$, mini-batch $B \subseteq [N]$, fairness parameter $\lambda \geq 0$, iteration number $R$.
2: **for** $t = 0, 1, \ldots, R$ **do**
3:     Draw a mini-batch $B$ of data points $\{(\mathbf{x}_i, s_i, y_i)\}_{i \in B}$
4:     Set $\boldsymbol{\theta}^{t+1} \leftarrow \boldsymbol{\theta}^t - \frac{\eta_\theta}{|B|} \sum_{i \in B} [\nabla_\theta \ell(\mathbf{x}_i, y_i; \boldsymbol{\theta}^t) + \lambda \nabla_\theta \psi_i(\boldsymbol{\theta}^t, W^t)]$.
5:     Set $W^{t+1} \leftarrow \Pi_{\mathcal{W}} \left( W^t + \frac{2\lambda \eta_w}{|B|} \sum_{i \in B} \left[ - W \widehat{\mathbf{y}}_i(\mathbf{x}_i; \boldsymbol{\theta}^t) \widehat{\mathbf{y}}_i^T(\mathbf{x}_i; \boldsymbol{\theta}^t) + P_s^{-1/2} \mathbf{s}_i \widehat{\mathbf{y}}_i^T(\mathbf{x}_i; \boldsymbol{\theta}^t) \right] \right)$
6: **end for**
7: Pick $\hat{t}$ uniformly at random from $\{1, \ldots, R\}$.
8: **Return:** $\boldsymbol{\theta}^{\hat{t}}$.

---

**Theorem 5.** *(Informal statement) Algorithm 1 converges to the set of $\epsilon$-first order stationary points of the Eq.* (FERMI obj.) *objective in* $O(\frac{1}{\epsilon^4})$ *iterations (stochastic gradient evaluations).*

The formal statement of this theorem can be found in Theorem 10 in Appendix D. A faster convergence rate of $O(\frac{1}{\epsilon^3})$ could be obtained by using the (more complicated) SREDA method of Luo et al. (2020) instead of SGDA to solve FERMI objective. We omit the details here. In the next section, we numerically evaluate the performance FERMI algorithm in several numerical experiments.

## 5 Numerical experiments

### 5.1 Binary classification and binary sensitive attribute

For our first set of experiments, we evaluate the fairness-accuracy tradeoffs of FERMI in binary classification problems with a binary sensitive attribute. This is a common setup, so we are able to compare against many existing baseline methods (Zafar et al., 2017; Feldman et al., 2015; Kamishima et al., 2011; Jiang et al., 2020; Hardt et al., 2016; Baharlouei et al., 2020; Rezaei et al., 2020; Donini et al., 2018; Cho et al., 2020b). We run experiments on three data sets: Adult, German Credit, and COMPAS. To implement FERMI, we train a logistic regression model (same model for all baselines) with an ERMI regularizer. Details about the datasets and experiments can be found in Appendix E.

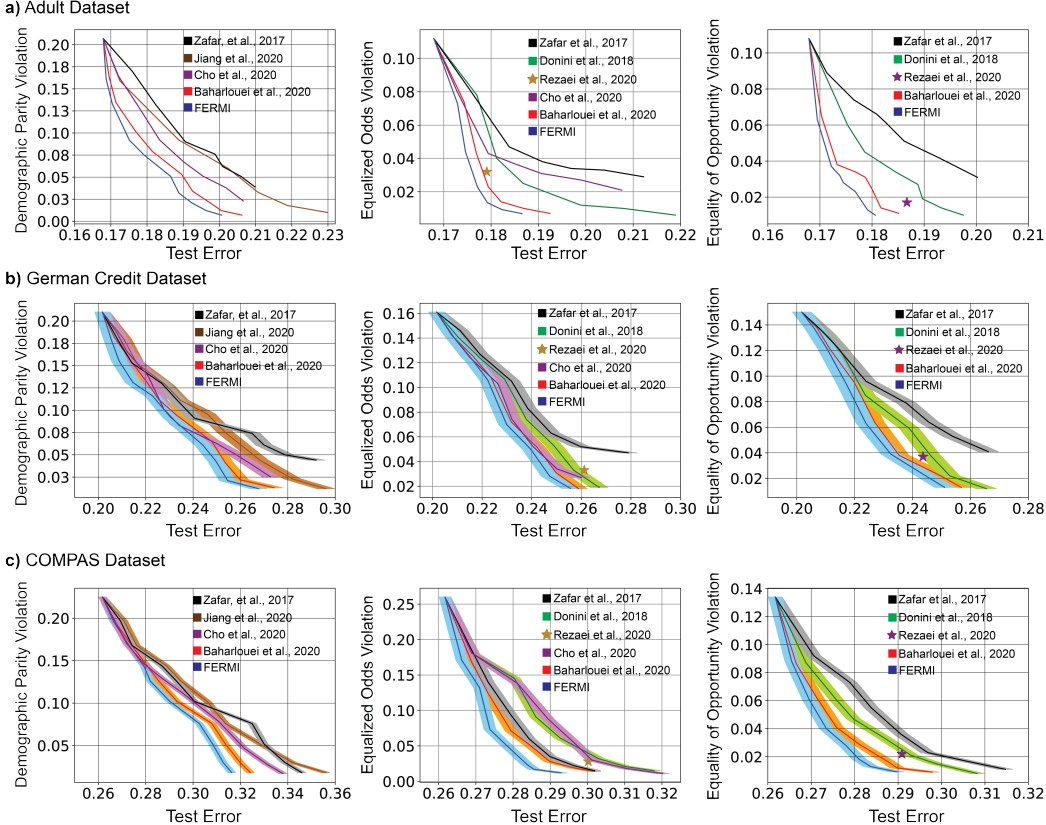

Figure 1: Binary classification with binary sensitive attribute using logistic regression. Tradeoff of fairness violation vs. test error for state-of-the-art fair classifiers on German Credit, Adult, and COMPAS datasets. FERMI offers the best fairness vs. accuracy tradeoff curve in all experiments against all baselines. Rezaei et al. (2020) only allow for a single output and do not yield a tradeoff curve. Further, the algorithms by Mary et al. (2019) and Baharlouei et al. (2020) are equivalent in this binary setting and shown by the red curve. FERMI, Mary et al. (2019) and Baharlouei et al. (2020) try to empirically solve the same risk function Eq. (FRMI obj.). However, the empirical formulation used by FERMI, Eq. (FERMI obj.) and its solver result in a better performance even-though we are using a full-batch for all baselines in this experiment.

In Fig. 1, we report the fairness violation vs. test error, for three notions of fairness: demographic parity, equalized odds, and equal opportunity. We have only included in-processing methods, which outperform pre-processing and post-processing methods. Complete experimental results are included in the appendix. We measure fairness violation through conditional demographic parity $L_\infty$ violation (Definition 9), conditional equal opportunity $L_\infty$ violation (Definition 10) and its generalization, conditional equalized odds violation. As can be seen, FERMI offers a fairness-accuracy tradeoff curve that dominates all existing state-of-the-art baselines in each experiment and with respect to each notion of fairness. This demonstrates the efficacy of having a strong regularizer such as ERMI: by enforcing small ERMI violation, our model simultaneously achieves small fairness violation with respect to these other notions which are upper bounded by ERMI.

It is noteworthy that the empirical objective function of Mary et al. (2019) and Baharlouei et al. (2020) is exactly the same in this setting, and their algorithms also coincide to the red curve in Fig. 1.[2] Additionally, like FERMI, they are trying to empirically solve Eq. (FRMI obj.), albeit using different estimation techniques, i.e., their empirical objective is different from Eq. (FERMI obj.). This demonstrates the effectiveness of our empirical formulation (FERMI obj.) – which is both unbiased and consistent whereas theirs is biased. It also shows the effectiveness of our solver (Algorithm 1) even-though we are using all baselines in full batch mode in this experiment. In the following experiments, we will demonstrate that using smaller batch sizes results in much more pronounced advantages of FERMI over these baselines.

---

[2]Exponential Rényi mutual information is equal to Rényi correlation for binary targets and/or binary sensitive attributes (see Theorem 2), which is the setting of all experiments in Sec. 5.1.

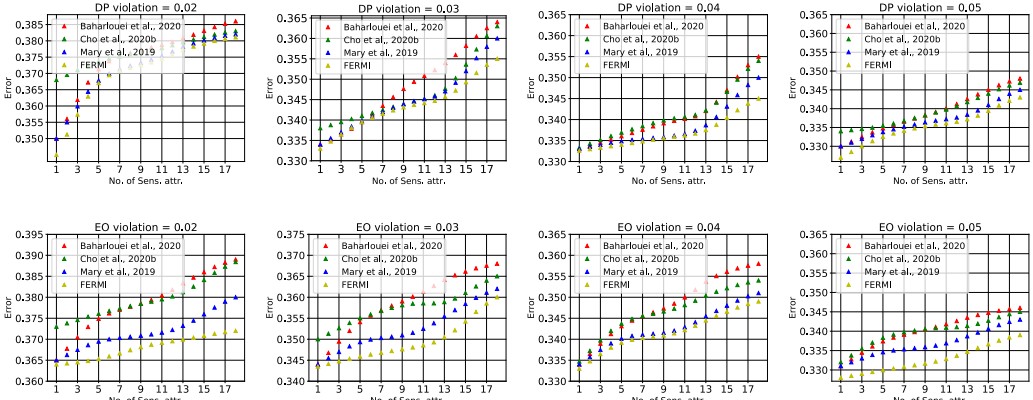

Figure 2: Comparison between FERMI, Mary et al. (2019), Baharlouei et al. (2020), and Cho et al. (2020b) on Communities dataset. (Mary et al., 2019) outperforms (Baharlouei et al., 2020; Cho et al., 2020b), which we believe could be attributed to the effectiveness of ERMI as a regularizer. FERMI outperforms Mary et al. (2019), which we attribute to our empirical formulation of ERMI and the effectiveness of its solver, given that we try to empirically solve the same risk function with different formulations.

## 5.2 Non-binary fair classification with a non-binary sensitive attribute

Next, we consider a non-binary classification problem with non-binary sensitive set. In this case, we consider the Communities and Crime dataset, which has 18 binary sensitive attributes in total, and we pick a subset of $1, 2, 3, \ldots, 18$ sensitive attributes out of those for our experiments, which corresponds to $|\mathcal{S}| \in \{2, 4, 8, \ldots, 2^{18}\}$. We discretize the target into three classes $\{\text{high}, \text{medium}, \text{low}\}$. The only baselines that we are aware of that can handle non-binary classification with non-binary sensitive attributes are (Mary et al., 2019), (Baharlouei et al., 2020), (Cho et al., 2020b), (Cho et al., 2020a), and (Zhang et al., 2018). We used the publicly available implementations of (Baharlouei et al., 2020) and (Cho et al., 2020b) and extended their binary classification algorithms to the non-binary setting.

The results are presented in Fig. 2, where we use conditional demographic parity $L_\infty$ violation (Definition 9) and conditional equal opportunity $L_\infty$ violation (Definition 10) as the fairness violation notions for the two experiments. For all baselines, test error increases as the number of sensitive attributes increases. As can be seen, compared to the baselines, FERMI offers the most favorable test error vs. fairness violation tradeoffs, particularly as the number of sensitive attributes increases and for the more stringent fairness violation levels, e.g., $0.02$.

## 5.3 Domain generalization through FERMI

In our last experiment, our goal is to showcase the efficacy of FERMI in stochastic optimization with neural network approximation. For this experiment, we consider the Color MNIST dataset (Li & Vasconcelos, 2019), where all 60,000 training MNIST digits are colored with different colors drawn from a class conditional Gaussian distribution with variance $\sigma$ around a certain average color for each digit, while the test set remains black and white. Li & Vasconcelos (2019) show that as $\sigma \to 0$, a convolutional network model overfits significantly to each digit's color on the training set, and achieves vanishing training accuracy. However, the learned representation does not generalize to the regular black and white test set, in absence of the spurious correlation between digits and color.

Conceptually, the goal of the classifier in this problem is to achieve high classification accuracy with predictions that are independent of the color of the digit. We view color as the sensitive attribute in this experiment, and apply fairness baselines for the demographic parity notion of fairness. One would expect that by promoting such independence through a fairness regularizer generalization would improve (i.e. lower test error on the black and white test set), at the cost of increased training error (on the colored training set). We compare against Mary et al. (2019), Baharlouei et al. (2020), and Cho et al. (2020b) as baselines in this experiment.

The results of this experiment are as illustrated in Fig. 3. The details about the dataset and experimental setup is provided in Appendix E. In the left panel, we see that with no regularization ($\lambda = 0$); the test error is around $80\%$. As $\lambda$ increases, all methods achieve smaller test error while training error increases. We also observe that FERMI offers the best test error in this setup. In the right panel, we observe that decreasing the batch size results in significantly worse generalization for all three

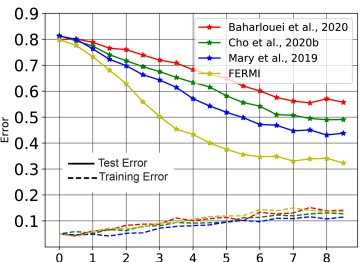 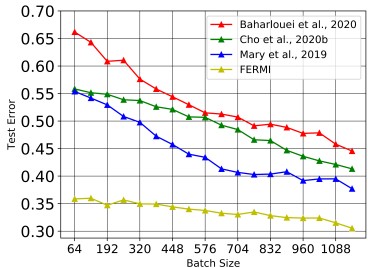

Figure 3: Domain generalization on Color MNIST ([Li & Vasconcelos, 2019](#)) using in-process fair algorithms for demographic parity. **Left panel:** The dashed line is the training error and the solid line is test error. As $\lambda$ increases, fairness regularization results in a learned representation that is less dependent on color; hence training error increases while test error decreases (all algorithms reach a plateau around $\lambda = 8$). We use $|B| = 512$ for all baselines. **Right panel:** We plot test error vs. batch size using an optimized value of $\lambda$ for each algorithm selected via a validation set. The performance of all baselines drops 10-20% as batch size becomes small whereas FERMI is relatively insensitive to batch size.

baselines considered (due to their biased estimators for the regularizer). However, the impact is much less on FERMI. In particular, the performance gap between FERMI and other baselines is more than 20% for $|B| = 64$. Finally, FERMI with minibatch size $|B| = 64$ still outperforms all other baselines with $|B| > 1,000$. Finally, notice that the test error achieved by FERMI when $\sigma = 0$ is $\sim 30\%$, as compared to more than $50\%$ obtained using REPAIR ([Li & Vasconcelos, 2019](#)) for $\sigma \leq 0.05$.

## 6 Discussion & concluding remarks

In this paper, we studied three variants of a notion of fairness violation, called exponential Rényi mutual information (ERMI), developed for demographic parity, equalized odds, and equal opportunity notions of fairness. We showed that ERMI is a strong fairness violation divergence providing upper bound guarantees on other popular violation divergences, namely Shannon mutual information, Rényi mutual information (Theorem 1), Pearson correlation, Rényi correlation (Theorem 2), and $L_q$ distance violation (Theorem 3).

We derived an unbiased estimator for ERMI (Lemma 1), based on which we formulated an empirical objective (FERMI obj.) for solving fair empirical risk minimization with ERMI regularization to balance performance and fairness. We provided a stochastic algorithm for solving FERMI (Algorithm 1) and proved its convergence (Theorem 5); for non-binary sensitive attributes, non-binary target variables, regardless of the batch size. From an experimental perspective, we showed that FERMI leads to better fairness-accuracy tradeoffs than all of the state-of-the-art baselines on a wide variety of binary and non-binary classification tasks (for demographic parity, equalized odds, and equal opportunity). We also showed that these benefits are particularly significant when the number of sensitive attributes grows or the batch size is small. In particular, we observed that FERMI consistently outperforms [Mary et al. (2019)](#) (which tries to empirically solve the same objective Eq. (FRMI obj.)) by up to 20% when the batch size is small, suggesting that the unbiasedness of the FERMI estimator is essential in achieving good empirical performance.

There are several possible explanations for the superior empirical performance of FERMI compared to baselines. One possible reason is that the objective function Eq. (FERMI obj.) is easier to optimize than the objectives of competing in-processing methods: ERMI is smooth; and in the discrete case, is equal to the trace of a matrix (see Theorem 7, appendix), which is easy to compute. Contrast this with the larger computational overhead of Rényi correlation used by [Baharlouei et al. (2020)](#), for example, which requires finding the second singular value of a matrix. Furthermore, the sample complexity of estimating Rényi mutual information of order 2 (and consequently that of ERMI) scales as $\Theta(\sqrt{|\mathcal{S}|})$ as compared to Shannon mutual information which scales as $\Theta(|\mathcal{S}|/\log|\mathcal{S}|)$ ([Acharya et al., 2014](#)). Moreover, the fact that ERMI is a stronger fairness violation seems to imply that FERMI would generalize well to other fairness notions, a hypothesis that is supported by our experimental results. Together, these facts suggest that ERMI serves as an efficient and easily optimizable proxy for these other fairness notions, making Eq. (FERMI obj.) a good surrogate objective to optimize for all three notions of fairness considered (demographic parity, equalized odds, and equal opportunity). We leave it as future work to rigorously understand which of these (or other) factors are most responsible for the favorable performance tradeoffs observed from FERMI.

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
