# Appendix

We provide a simple table of contents below for easier navigation of the appendix.

**CONTENTS**

# A  Existing notions of fairness

Let $(Y, \widehat{Y}, \mathcal{A}, S)$ denote the true target, predicted target, the advantaged outcome class, and the sensitive attribute, respectively. We review three major notions of fairness.

**Definition 6** (demographic parity (Dwork et al., 2012))**.** *We say that a learning machine satisfies demographic parity if $\widehat{Y}$ is independent of $S$.*

**Definition 7** (equalized odds (Hardt et al., 2016))**.** *We say that a learning machine satisfies equalized odds, if $\widehat{Y}$ is conditionally independent of $S$ given $Y$.*

**Definition 8** (equal opportunity (Hardt et al., 2016))**.** *We say that a learning machine satisfies equal opportunity with respect to $\mathcal{A}$, if $\widehat{Y}$ is conditionally independent of $S$ given $Y = y$ for all $y \in \mathcal{A}$.*

Notice that the equal opportunity as defined here generalizes the definition in (Hardt et al., 2016). It recovers equalized odds if $\mathcal{A} = \mathcal{Y}$, and it recovers equal opportunity of (Hardt et al., 2016) for $\mathcal{A} = \{1\}$ in binary classification.

 # B   Properties and special cases of ERMI

Notice that ERMI is in fact the $\chi^2$-divergence between the conditional joint distribution, $p_{\widehat{Y},S}$, and the Kronecker product of conditional marginals, $p_{\widehat{Y}} \otimes p_S$, where the conditioning is on $Z \in \mathcal{Z}$. Further, $\chi^2$-divergence is an $f$-divergence with $f(t) = (t-1)^2$. See (Csiszár & Shields, 2004, Section 4) for a discussion. As an immediate result of this observation and well-known properties of $f$-divergences, we can state the following property of ERMI:

**Remark 6.** $D_R(\widehat{Y}; S | Z \in \mathcal{Z}) \geq 0$ *with equality if and only if for all* $z \in \mathcal{Z}$, $\widehat{Y}$ *and* $S$ *are conditionally independent given* $Z = z$.

To further clarify the definition of ERMI, especially as it relates to demographic parity, equalized odds, and equal opportunity, we will unravel the definition explicitly in a few special cases.

First, let $Z = 0$ and $\mathcal{Z} = \{0\}$. In this case, $Z \in \mathcal{Z}$ trivially holds, and conditioning on $Z$ has no effect, resulting in:

$$
\begin{aligned}
D_R(\widehat{Y}; S) := D_R(\widehat{Y}; S | Z \in \mathcal{Z})\Big|_{Z=0, \mathcal{Z}=\{0\}} \\
= \mathbb{E}_{\widehat{Y},S}\left\{ \frac{p_{\widehat{Y},S}(\widehat{Y}, S)}{p_{\widehat{Y}}(\widehat{Y}) p_S(S)} \right\} - 1 \\
= \sum_{s \in \mathcal{S}} \int_{\widehat{y} \in \mathcal{Y}} \frac{p_{\widehat{Y},S}(\widehat{y}, s) - p_{\widehat{Y}}(\widehat{y}) p_S(s)}{p_{\widehat{Y}}(\widehat{y}) p_S(s)} p_{\widehat{Y},S}(\widehat{y}, s) d\widehat{y}.
\end{aligned}
\tag{5}
$$

$D_R(\widehat{Y}; S)$ is the notion of ERMI that should be used when the desired notion of fairness is demographic parity. In particular, $D_R(\widehat{Y}; S) = 0$ implies that $\chi^2$ divergence between $p_{\widehat{Y},S}$, and the Kronecker product of marginals, $p_{\widehat{Y}} \otimes p_S$ is zero. This in turn implies that $\widehat{Y}$ and $S$ are independent, which is the definition of demographic parity. We note that when $\widehat{Y}$ and $S$ are discrete, this special case ($Z = 0$ and $\mathcal{Z} = \{0\}$) of ERMI is referred to as $\chi^2$-information by Calmon et al. (2017a).

Next, we consider $Z = Y$ and $\mathcal{Z} = \mathcal{Y}$. In this case, $Z \in \mathcal{Z}$ is trivially satisfied, and hence,

$$
\begin{aligned}
D_R(\widehat{Y}; S | Y) := D_R(\widehat{Y}; S | Z \in \mathcal{Z})\Big|_{Z=Y, \mathcal{Z}=\mathcal{Y}} \\
= \mathbb{E}_{Y,\widehat{Y},S}\left\{ \frac{p_{\widehat{Y},S|Y}(\widehat{Y}, S|Y)}{p_{\widehat{Y}|Y}(\widehat{Y}|Y) p_{S|Y}(S|Y)} \right\} - 1 \\
= \sum_{s \in \mathcal{S}} \int_{y \in \mathcal{Y}} \int_{\widehat{y} \in \mathcal{Y}} \frac{p_{\widehat{Y},S|Y}(\widehat{y}, s|y) - p_{\widehat{Y}|Y}(\widehat{y}|y) p_{S|Y}(s|y)}{p_{\widehat{Y}|Y}(\widehat{y}|y) p_{S|Y}(s|y)} p_{Y,\widehat{Y},S}(y, \widehat{y}, s) d\widehat{y} dy \\
= \sum_{s \in \mathcal{S}} \int_{y \in \mathcal{Y}} \int_{\widehat{y} \in \mathcal{Y}} \frac{p_{\widehat{Y},S|Y}(\widehat{y}, s|y)^2}{p_{\widehat{Y}|Y}(\widehat{y}|y) p_{S|Y}(s|y)} p_Y(y) d\widehat{y} dy - 1.
\end{aligned}
\tag{6}
$$

$D_R(\widehat{Y}; S | Y)$ should be used when the desired notion of fairness is equalized odds. In particular, $D_R(\widehat{Y}; S | Y) = 0$ directly implies the conditional independence of $\widehat{Y}$ and $S$ given $Y$.

Finally, we consider $Z = Y$ and $\mathcal{Z} = \mathcal{A}$. In this case, we have

$$
\begin{aligned}
D_R^{\mathcal{A}}(\widehat{Y}; S | Y) := D_R(\widehat{Y}; S | Z \in \mathcal{Z})\Big|_{Z=Y, \mathcal{Z}=\mathcal{A}} \\
= \mathbb{E}_{Y,\widehat{Y},S}\left\{ \frac{p_{\widehat{Y},S|Y}(\widehat{Y}, S|Y)}{p_{\widehat{Y}|Y}(\widehat{Y}|Y) p_{S|Y}(S|Y)} \bigg| Y \in \mathcal{A} \right\} - 1 \\
= \sum_{s \in \mathcal{S}} \int_{y \in \mathcal{A}} \int_{\widehat{y} \in \mathcal{Y}} \frac{p_{\widehat{Y},S|Y}(\widehat{y}, s|y) - p_{\widehat{Y}|Y}(\widehat{y}|y) p_{S|Y}(s|y)}{p_{\widehat{Y}|Y}(\widehat{y}|y) p_{S|Y}(s|y)} p_Y^{\mathcal{A}}(y) d\widehat{y} dy \\
= \sum_{s \in \mathcal{S}} \int_{y \in \mathcal{A}} \int_{\widehat{y} \in \mathcal{Y}} \frac{p_{\widehat{Y},S|Y}(\widehat{y}, s|y)^2}{p_{\widehat{Y}|Y}(\widehat{y}|y) p_{S|Y}(s|y)} p_{\widehat{Y},S|Y}(\widehat{y}, s|y) p_Y^{\mathcal{A}}(y) d\widehat{y} dy - 1,
\end{aligned}
\tag{7}
$$

where

$$p_Y^{\mathcal{A}}(y) := \frac{p_Y(y)}{\int_{y' \in \mathcal{A}} p_Y(y') dy'}. \tag{8}$$

This notion is what should be used when the desired notion of fairness is equal opportunity. This can be further simplified when the advantaged class is a singleton (which is the case in binary classification). If $Z = Y$ and $\mathcal{Z} = \{y\}$, then

$$
\begin{aligned}
D_R(\widehat{Y}; S | Y = y) &:= D_R^{\{y\}}(\widehat{Y}; S | Y) \\
&= \sum_{s \in \mathcal{S}} \int_{\widehat{y} \in \mathcal{Y}} \frac{p_{\widehat{Y}, S | Y}(\widehat{y}, s | y) - p_{\widehat{Y} | Y}(\widehat{y} | y) p_{S | Y}(s | y)}{p_{\widehat{Y} | Y}(\widehat{y} | y) p_{S | Y}(s | y)} p_{\widehat{Y}, S | Y}(\widehat{y}, s | y) d\widehat{y} \\
&= \sum_{s \in \mathcal{S}} \int_{\widehat{y} \in \mathcal{Y}} \frac{p_{\widehat{Y}, S | Y}(\widehat{y}, s | y)^2}{p_{\widehat{Y} | Y}(\widehat{y} | y) p_{S | Y}(s | y)} d\widehat{y} - 1.
\end{aligned}
\tag{9}
$$

Finally, we note that we use the notation $D_R(\widehat{Y}; S | Y)$ and $D_R(\widehat{Y}; S | Y = y)$ to be consistent with the definition of conditional mutual information in (Cover & Thomas, 1991).

# C   Relations between ERMI and other fairness violation notions

*Proof of Theorem 1.* We proceed to prove all the (in)equalities one by one:

- $0 \leq I_S(\widehat{Y}; S|Z \in \mathcal{Z})$. This is well known and the proof can be found in any information theory textbook (Cover & Thomas, 1991).

- $I_1(\widehat{Y}; S|Z \in \mathcal{Z}) \leq I_2(\widehat{Y}; S|Z \in \mathcal{Z})$. This is a known property of Rényi mutual information, but we provide a proof for completeness in Lemma 2.

- $I_2(\widehat{Y}; S|Z \in \mathcal{Z}) \leq e^{I_2(\widehat{Y}; S|Z \in \mathcal{Z})} - 1$. This follows from the fact that $x \leq e^x - 1$.

- $e^{I_2(\widehat{Y}; S)|Z \in \mathcal{Z}} - 1 = D_R(\widehat{Y}; S|Z \in \mathcal{Z})$. This follows from simple algebraic manipulation.

$\square$

**Lemma 2.** *Let $\widehat{Y}, S, Z$ be discrete or continuous random variables. Then:*

(a) *For any $\alpha, \beta \in [1, \infty]$, $I_\beta(\widehat{Y}; S|Z \in \mathcal{Z}) \geq I_\alpha(\widehat{Y}; S|Z \in \mathcal{Z})$ if $\beta > \alpha$.*

(b) $\lim_{\alpha \to 1^+} I_\alpha(\widehat{Y}; S|Z \in \mathcal{Z}) = I_1(\widehat{Y}; S) := \mathbb{E}_Z \left\{ D_{KL}(p_{\widehat{Y}, S|Z} || p_{\widehat{Y}|Z} \otimes p_{S|Z}) \Big| Z \in \mathcal{Z} \right\}$,
*where $I_1(\cdot; \cdot)$ denotes the Shannon mutual information and $D_{KL}$ is Kullback–Leibler divergence (relative entropy).*

(c) *For all $\alpha \in [1, \infty]$, $I_\alpha(\widehat{Y}; S|Z \in \mathcal{Z}) \geq 0$ with equality if and only if for all $z \in \mathcal{Z}$, $\widehat{Y}$ and $S$ are conditionally independent given $z$.*

*Proof. (a)* First assume $0 < \alpha < \beta < \infty$ and that $\alpha, \beta \neq 1$. Define $a = \alpha - 1$, and $b = \beta - 1$. Then the function $\phi(t) = t^{b/a}$ is convex for all $t \geq 0$, so by Jensen's inequality we have:

$$\frac{1}{b} \log \left( \mathbb{E} \left\{ \left( \frac{p(\widehat{Y}, S|Z)}{p(\widehat{Y}|Z)p(S|Z)} \right)^b \Bigg| Z \in \mathcal{Z} \right\} \right) \geq \frac{1}{b} \log \left( \mathbb{E} \left\{ \left( \frac{p(\widehat{Y}, S|Z)}{p(\widehat{Y}|Z)p(S|Z)} \right)^a \Bigg| Z \in \mathcal{Z} \right\}^{b/a} \right)$$
$$= \frac{1}{a} \log \left( \mathbb{E} \left\{ \left( \frac{p(\widehat{Y}, S|Z)}{p(\widehat{Y}|Z)p(S|Z)} \right)^a \Bigg| Z \in \mathcal{Z} \right\} \right). \tag{10}$$

Now suppose $\alpha = 1$. Then by the monotonicity for $\alpha \neq 1$ proved above, we have $I_1(\widehat{Y}; S) = \lim_{\alpha \to 1^-} I_\alpha(\widehat{Y}; S) = \sup_{\alpha \in (0,1)} I_\alpha(\widehat{Y}; S) \leq \inf_{\alpha > 1} I_\alpha(\widehat{Y}; S)$. Also, $I_\infty(\widehat{Y}; S) = \lim_{\alpha \to \infty} I_\alpha(\widehat{Y}; S) = \sup_{\alpha > 0} I_\alpha(\widehat{Y}; S)$.

*(b)* This is a standard property of the cumulant generating function (see (Dembo & Zeitouni, 2009)).

*(c)* It is straightforward to observe that independence implies that Rényi mutual information vanishes. On the other hand, if Rényi mutual information vanishes, then part (a) implies that Shannon mutual information also vanishes, which implies the desired conditional independence. $\square$

*Proof of Theorem 2.* The proof is completed using the following pieces.

- $0 \leq |\rho(\widehat{Y}, S|Z \in \mathcal{Z})| \leq \rho_R(\widehat{Y}, S|Z \in \mathcal{Z})$. This is obvious from the definition of $\rho_R(\widehat{Y}, S|Z \in \mathcal{Z})$.

- $\rho_R(\widehat{Y}, S|Z \in \mathcal{Z}) \leq D_R(\widehat{Y}; S|Z \in \mathcal{Z})$. This follows from Theorem 7.

- Notice that if $|\mathcal{S}| = 2$, Theorem 7 implies that $D_R(\widehat{Y}; S|Z \in \mathcal{Z}) = \rho_R(\widehat{Y}, S|Z \in \mathcal{Z})$.

$\square$

**Theorem 7.** *Suppose that $\mathcal{S} = [k]$. Let the $k \times k$ matrix $P$ be defined as $P = \{P_{ij}\}_{i,j \in [k] \times [k]}$, where*

$$P_{ij} := \frac{1}{\sqrt{p_S(i)p_S(j)}} \int_{y \in \mathcal{Y}} \left( \frac{p_{\widehat{Y},S}(y,i)p_{\widehat{Y},S}(y,j)}{p_{\widehat{Y}}(y)} \right) dy. \tag{11}$$

*Let $1 = \sigma_1 \geq \sigma_2 \geq \ldots \geq \sigma_k \geq 0$ be the eigenvalues of $P$. Then,*

$$\rho_R(\widehat{Y}, S) = \sigma_2, \tag{12}$$

$$D_R(\widehat{Y}; S) = \text{Tr}(P) - 1 = \sum_{i=2}^{k} \sigma_i. \tag{13}$$

*Proof.* Eq. (12) is proved in (Witsenhausen, 1975, Section 3). To prove Eq. (13), notice that

$$\text{Tr}(P) = \sum_{i \in [k]} P_{ii}$$

$$= \sum_{i \in [k]} \frac{1}{p_S(i)} \int_{y \in \mathcal{Y}} \left( \frac{p_{\widehat{Y},S}(y,i)^2}{p_{\widehat{Y}}(y)} \right) dy$$

$$= E_{\widehat{Y},S} \left\{ \left( \frac{p_{\widehat{Y},S}(\widehat{Y}, S)}{p_{\widehat{Y}}(\widehat{Y})p_S(S)} \right) \right\}$$

$$= 1 + D_R(\widehat{Y}; S),$$

which completes the proof. $\qquad\square$

*Proof of Theorem 3.* It suffices to prove the inequality for $L_1$, as $L_q$ is bounded above by $L_1$ for all $q \geq 1$. The proof for the case where $Z = 0$ and $\mathcal{Z} = \{0\}$ follows from the following set of inequalities:

$$L_1(\widehat{Y}, S | Z \in \mathcal{Z}) = \sum_{s \in \mathcal{S}} \int_{y \in \mathcal{Y}} \left| p_{\widehat{Y},S}(y,s) - p_{\widehat{Y}}(y)p_S(s) \right| dy \tag{14}$$

$$= \sum_{s \in \mathcal{S}} \int_{y \in \mathcal{Y}} \sqrt{p_{\widehat{Y}}(y)p_S(s)} \frac{\left| p_{\widehat{Y},S}(y,s) - p_{\widehat{Y}}(y)p_S(s) \right|}{\sqrt{p_{\widehat{Y}}(y)p_S(s)}} dy \tag{15}$$

$$\leq \sqrt{\left( \sum_{s \in \mathcal{S}} \int_{y \in \mathcal{Y}} p_{\widehat{Y}}(y)p_S(s) dy \right) \left( \sum_{s \in \mathcal{S}} \int_{y \in \mathcal{Y}} \left( \frac{(p_{\widehat{Y},S}(y,s) - p_{\widehat{Y}}(y)p_S(s))^2}{p_{\widehat{Y}}(y)p_S(s)} \right) \right)} \tag{16}$$

$$\leq \sqrt{\sum_{s \in \mathcal{S}} \int_{y \in \mathcal{Y}} \left( \frac{(p_{\widehat{Y},S}(y,s) - p_{\widehat{Y}}(y)p_S(s))^2}{p_{\widehat{Y}}(y)p_S(s)} \right) dy} \tag{17}$$

$$= \sqrt{D_R(\widehat{Y}; S)}, \tag{18}$$

where Eq. (16) follows from Cauchy-Schwarz inequality, and Eq. (18) follows from Lemma 3. The extension to general $Z$ and $\mathcal{Z}$ is immediate by observing that $\rho(\widehat{Y}, S | Z \in \mathcal{Z}) = \mathbb{E}_Z \left[ \rho(\widehat{Y}, S | Z) \Big| Z \in \mathcal{Z} \right]$, $\rho_R(\widehat{Y}, S | Z \in \mathcal{Z}) = \mathbb{E}_Z \left[ \rho_R(\widehat{Y}, S | Z) \Big| Z \in \mathcal{Z} \right]$, and $D_R(\widehat{Y}, S | Z \in \mathcal{Z}) = \mathbb{E}_Z \left[ D_R(\widehat{Y}, S | Z) \Big| Z \in \mathcal{Z} \right]$.

$\qquad\square$

**Lemma 3.** *We have*

$$D_R(\widehat{Y}; S) = \sum_{s \in \mathcal{S}} \int_{y \in \mathcal{Y}} \left( \frac{(p_{\widehat{Y},S}(y,s) - p_{\widehat{Y}}(y)p_S(s))^2}{p_{\widehat{Y}}(y)p_S(s)} \right) dy. \tag{19}$$

*Proof.* The proof follows from the following set of identities:

$$\sum_{s\in\mathcal{S}}\int_{y\in\mathcal{Y}}\left(\frac{(p_{\widehat{Y},S}(y,s)-p_{\widehat{Y}}(y)p_S(s))^2}{p_{\widehat{Y}}(y)p_S(s)}\right)dy = \sum_{s\in\mathcal{S}}\int_{y\in\mathcal{Y}}\frac{(p_{\widehat{Y},S}(y,s))^2}{p_{\widehat{Y}}(y)p_S(s)}dy$$

$$- 2\sum_{s\in\mathcal{S}}\int_{y\in\mathcal{Y}}p_{\widehat{Y},S}(y,s)dy$$

$$+ \sum_{s\in\mathcal{S}}\int_{y\in\mathcal{Y}}p_{\widehat{Y}}(y)p_S(s)dy \qquad (20)$$

$$= E\left\{\frac{p_{\widehat{Y},S}(\widehat{Y},S)}{p_{\widehat{Y}}(\widehat{Y})p_S(S)}\right\} - 1 \qquad (21)$$

$$= D_R(\widehat{Y};S). \qquad (22)$$

$\square$

Next, we present some alternative fairness definitions and show that they are also upper bounded by ERMI.

**Definition 9** (conditional demographic parity $L_\infty$ violation)**.** *Given a predictor $\widehat{Y}$ supported on $\mathcal{Y}$ and a discrete sensitive attribute $S$ supported on a finite set $\mathcal{S}$, we define the conditional demographic parity violation by:*

$$\widetilde{dp}(\widehat{Y}|S) := \sup_{\widehat{y}\in\mathcal{Y}}\max_{s\in\mathcal{S}}\left|p_{\widehat{Y}|S}(\widehat{y}|s)-p_{\widehat{Y}}(\widehat{y})\right|. \qquad (23)$$

First, we show that $\widetilde{dp}(\widehat{Y}|S)$ is a reasonable notion of fairness violation.

**Lemma 4.** $\widetilde{dp}(\widehat{Y}|S) = 0$ *iff (if and only if) $\widehat{Y}$ and $S$ are independent.*

*Proof.* By definition, $\widetilde{dp}(\widehat{Y}|S) = 0$ iff for all $\widehat{y}\in\mathcal{Y}, s\in\mathcal{S}, p_{\widehat{Y},S}(\widehat{y}|s) = p_{\widehat{Y}}(\widehat{y})$ iff $\widehat{Y}$ and $S$ are independent (since we always assume $p(s) > 0$ for all $s\in\mathcal{S}$). $\square$

**Theorem 8** (ERMI is stronger than conditional demographic parity $L_\infty$ violation)**.** *Let $\widehat{Y}$ be a discrete or continuous random variable supported on $\mathcal{Y}$, and $S$ be a discrete random variable supported on a finite set $\mathcal{S}$. Denote $p_S^{\min} := \min_{s\in\mathcal{S}} p_S(s) > 0$. Then,*

$$0 \le \widetilde{dp}(\widehat{Y}|S) \le \frac{1}{p_S^{\min}}\sqrt{D_R(\widehat{Y};S)}. \qquad (24)$$

*Proof.* The proof follows from the following set of (in)equalities:

$$\left(\widetilde{dp}(\widehat{Y}|S)\right)^2 = \sup_{\widehat{y}\in\mathcal{Y}}\max_{s\in\mathcal{S}}\left(p_{\widehat{Y}|S}(\widehat{y}|s)-p_{\widehat{Y}}(\widehat{y})\right)^2 \qquad (25)$$

$$\le \frac{1}{(p_S^{\min})^2}\sup_{\widehat{y}\in\mathcal{Y}}\max_{s\in\mathcal{S}}\left(p_{\widehat{Y},S}(\widehat{y},s)-p_{\widehat{Y}}(\widehat{y})p_S(s)\right)^2 \qquad (26)$$

$$\le \frac{1}{(p_S^{\min})^2}\int_{\widehat{y}\in\mathcal{Y}}\sum_{s\in\mathcal{S}}\left(p_{\widehat{Y},S}(\widehat{y},s)-p_{\widehat{Y}}(\widehat{y})p_S(s)\right)^2 \qquad (27)$$

$$= \frac{1}{(p_S^{\min})^2}D_R(\widehat{Y};S), \qquad (28)$$

where Eq. (28) follows from Theorem 3. $\square$

**Definition 10** (conditional equal opportunity $L_\infty$ violation (Hardt et al., 2016))**.** *Let $Y, \widehat{Y}$ take values in $\mathcal{Y}$ and let $\mathcal{A}\subseteq\mathcal{Y}$ be a compact subset denoting the advantaged outcomes (For example, the decision "to interview" an individual or classify an individual as a "low risk" for financial purposes).*

We define the conditional equal opportunity $L_\infty$ violation of $\widehat{Y}$ with respect to the sensitive attribute $S$ and the advantaged outcome $\mathcal{A}$ by

$$\widetilde{eo}(\widehat{Y}|S, Y \in \mathcal{A}) := \mathbb{E}_Y \left\{ \sup_{\widehat{y} \in \mathcal{Y}} \max_{s \in \mathcal{S}} \left| p_{\widehat{Y}, S|Y}(\widehat{y}|s, Y) - p_{\widehat{Y}|Y}(\widehat{y}|Y) \right| \middle| Y \in \mathcal{A} \right\}. \qquad (29)$$

**Theorem 9** (ERMI is stronger than conditional equal opportunity $L_\infty$ violation)**.** *Let $\widehat{Y}$, $Y$, be discrete or continuous random variables supported on $\mathcal{Y}$, and let $S$ be a discrete random variable supported on a finite set $\mathcal{S}$. Let $\mathcal{A} \subseteq \mathcal{Y}$ be a compact subset of $\mathcal{Y}$.*

*Denote $p_{S|\mathcal{A}}^{\min} = \min_{s \in \mathcal{S}, y \in \mathcal{A}} p_{S|Y}(s|y)$. Then,*

$$0 \leq \widetilde{eo}(\widehat{Y}|S, Y \in \mathcal{A}) \leq \frac{1}{p_{S|\mathcal{A}}^{\min}} \sqrt{D_R(\widehat{Y}; S|Y \in \mathcal{A})}. \qquad (30)$$

*Proof.* Notice that the same proof for Theorem 8 would give that for all $y \in \mathcal{A}$:

$$0 \leq \sup_{\widehat{y} \in \mathcal{Y}} \max_{s \in \mathcal{S}} \left| p_{\widehat{Y}, S|Y}(\widehat{y}|s, y) - p_{\widehat{Y}|Y}(\widehat{y}|y) \right| := \widetilde{eo}(\widehat{Y}|S, Y = y)$$

$$\leq \frac{1}{p_{S|y}^{\min}(y)} \sqrt{D_R(\widehat{Y}; S|Y = y)}$$

$$\leq \frac{1}{p_{S|\mathcal{C}}^{\min}} \sqrt{D_R(\widehat{Y}; S|Y = y)}.$$

Hence,

$$\widetilde{eo}(\widehat{Y}|S, Y \in \mathcal{A}) = \mathbb{E}_Y \left\{ \widetilde{eo}(\widehat{Y}|S, Y) \middle| Y \in \mathcal{A} \right\}$$

$$\leq \frac{1}{p_{S|\mathcal{A}}^{\min}} \mathbb{E}_Y \left\{ \sqrt{D_R(\widehat{Y}; S|Y)} \middle| Y \in \mathcal{A} \right\}$$

$$\leq \frac{1}{p_{S|\mathcal{A}}^{\min}} \sqrt{\mathbb{E}_Y \left\{ D_R(\widehat{Y}; S|Y) \middle| Y \in \mathcal{A} \right\}}$$

$$= \frac{1}{p_{S|\mathcal{A}}^{\min}} \sqrt{D_R(\widehat{Y}; S|Y \in \mathcal{A})},$$

where the last inequality follows from Jensen's inequality. This completes the proof. $\qquad \square$

 # D   FERMI: objective and algorithm

*Proof of Theorem 4.* Let $W^* \in \arg\max_{W \in \mathbb{R}^{k \times m}} -\operatorname{Tr}(W P_{\widehat{y}} W^T) + 2\operatorname{Tr}(W P_{\widehat{y},s} P_s^{-1/2})$. We will compute $W^*$ and plug it in the RHS of Eq. (4) to show the equality in Eq. (4). Setting the derivative of the expression on the RHS equal to zero leads to:

$$-2W P_{\widehat{y}} + 2 P_s^{-1/2} P_{\widehat{y},s}^T = 0 \implies W^* = P_{\widehat{y}}^{-1} P_{\widehat{y},s}^T P_s^{-1/2}.$$

Plugging this expression for $W^*$, we have

$$\max_{W \in \mathbb{R}^{k \times m}} -\operatorname{Tr}(W P_{\widehat{y}} W^T) + 2\operatorname{Tr}(W P_{\widehat{y},s} P_s^{-1/2})$$

$$= -\operatorname{Tr}(P_s^{-1/2} P_{\widehat{y},s}^T P_{\widehat{y}}^{-1} P_{\widehat{y}} P_{\widehat{y}}^{-1} P_s^{-1/2}) + 2\operatorname{Tr}(P_s^{-1/2} P_{\widehat{y},s}^T P_{\widehat{y}}^{-1} P_{\widehat{y}} P_{\widehat{y}}^{-1} P_s^{-1/2})$$

$$= \operatorname{Tr}(P_s^{-1/2} P_{\widehat{y},s}^T P_{\widehat{y}}^{-1} P_{\widehat{y},s} P_s^{-1/2})$$

$$= \operatorname{Tr}(P_s^{-1} P_{\widehat{y},s}^T P_{\widehat{y}}^{-1} P_{\widehat{y},s}).$$

Writing out the matrix multiplication explicitly in the last expression, we have

$$P_s^{-1} P_{\widehat{y},s}^T P_{\widehat{y}}^{-1} P_{\widehat{y},s} = U V^T,$$

where $U_{i,j} = \widehat{p}_S(i)^{-1} \widehat{p}_{\widehat{Y},S}(j,i)$ and $V_{i,j} = \hat{p}_{\widehat{Y}}(j)^{-1} \hat{p}_{\widehat{Y},S}(j,i)$, for $i \in [k], j \in [m]$. Hence

$$\max_{W \in \mathbb{R}^{k \times m}} -\operatorname{Tr}(W P_{\widehat{y}} W^T) + 2\operatorname{Tr}(W P_{\widehat{y},s} P_s^{-1/2}) = \operatorname{Tr}(U V^T)$$

$$= \sum_{i \in [k]} \sum_{j \in [m]} \frac{p_{\widehat{Y},S}(j,i)^2}{p_S(i) p_{\widehat{Y}}(j)}$$

$$= D_R(\widehat{Y}; S),$$

which completes the proof. $\qquad\square$

Next, we move to the statement and proof of the precise version of Theorem 5. We first recall some basic definitions:

**Definition 11.** *A function $f$ is $\beta$-smooth if for all $\mathbf{u}, \mathbf{u}'$, we have $\|\nabla f(\mathbf{u}) - \nabla f(\mathbf{u})\| \leq \beta \|\mathbf{u} - \mathbf{u}'\|$.*

**Definition 12.** *A point $\boldsymbol{\theta}$ is an $\epsilon$-stationary point of a differentiable function $\Phi$ if $\|\nabla \Phi(\boldsymbol{\theta})\| \leq \epsilon$.*

**Assumption 1.**         • $\ell$ *is twice differentiable, $L_\ell$-Lipscthiz, and $\beta_\ell$-smooth in $\boldsymbol{\theta}$.*

- $\|\nabla_\theta P_{\hat{y}}\|_2 := \|\nabla_\theta \operatorname{vec}(P_{\widehat{y}})\|_2 \leq L_y$ *and* $\max_{l \in [m]} \|\nabla_\theta ((P_{\widehat{y}})_{l,l})\|_2 \leq \widetilde{L}_y$

- $\max_{l \in [m]} \|\nabla_{\theta\theta}^2 (P_{\widehat{y}})_{l,l}\|_2 \leq \beta_y.$

- $\|\nabla_\theta P_{\hat{y},s}^T\|_2 := \|\nabla_\theta \operatorname{vec}(P_{\widehat{y},s}^T)\|_2 \leq L_{ys}$ *and* $\max_{l \in [m], j \in [k]} \|\nabla_\theta ((P_{\widehat{y},s})_{l,m})\|_2 \leq \widetilde{L}_{ys}$

- $\max_{l \in [m], j \in [k]} \|\nabla_{\theta\theta}^2 (P_{\widehat{y},s})_{l,j}\|_2 \leq \beta_{y,s}.$

**Theorem 10** (Precise statement of Theorem 5). *Denote*

$$f(\boldsymbol{\theta}, W) = \frac{1}{N} \sum_{i \in [N]} \ell(\mathbf{x}_i, y_i; \boldsymbol{\theta}) + \lambda \left( -\operatorname{Tr}(W P_{\widehat{y}} W^T) + 2\operatorname{Tr}(W P_{\widehat{y},s} P_s^{-1/2}) - 1 \right).$$

*Set $\mathcal{W} := B_F(0, 2D) \subset \mathbb{R}^{k \times m}$ (Frobenius norm ball of radius $2D$), $D := \frac{\sqrt{mk}}{\hat{p}_{\widehat{y}}^{\min} \sqrt{\hat{p}_s^{\min}}}$. Denote $\Delta_\Phi := \Phi(\theta_0) - \min_\theta \Phi(\theta)$, where $\Phi(\theta) := \max_{W \in \mathcal{W}} f(\boldsymbol{\theta}, W)$. In Algorithm 1, choose the step-sizes as $\eta_\theta = \Theta(1/\kappa^2 \beta)$ and $\eta_W = \Theta(1/\beta)$ and mini-batch size as $M = \Theta\left(\max\left\{1, \kappa\sigma^2 \epsilon^{-2}\right\}\right)$. Then under Assumption 1, the iteration complexity of Algorithm 1 to return an $\epsilon$-stationary point of $f$ is bounded by*

$$\mathcal{O}\left( \frac{\kappa^2 \beta \Delta_\Phi + \kappa \beta^2 D^2}{\epsilon^2} \right),$$

627 *which gives the total stochastic gradient complexity of*

$$\mathcal{O}\left(\left(\frac{\kappa^2\beta\Delta_\Phi + \kappa\beta^2 D^2}{\epsilon^2}\right)\max\left\{1, \kappa\sigma^2\epsilon^{-2}\right\}\right),$$

628 *where*

$$\beta = \beta_l + 8\lambda D^2\beta_y + 4\lambda\frac{1}{\sqrt{\hat{p}_s^{\min}}}\left(\sqrt{m}k^{3/2}D\beta_{ys}\right) + 2\lambda + 4\lambda\left(DL_y + \frac{L_{ys}}{\sqrt{\hat{p}_s^{\min}}}\right),$$

$$\mu = 2\lambda\hat{p}_{\hat{y}}^{\min},$$

$$\kappa = \beta/\mu,$$

$$\sigma^2 = 2\left(L_\ell + 2\lambda\widetilde{L}_y D^2 + 4\lambda\frac{D}{\sqrt{\hat{p}_s^{\min}}}\sqrt{m}k\widetilde{L}_{ys}\right)^2 + 2\left(2\lambda D + 2(\hat{p}_s^{\min})^{-1/2}\sqrt{m}k\right)^2.$$

629 The theorem follows from Theorem 4.5 in (Lin et al., 2020) combined with the following technical
630 lemmas. We assume Assumption 1 holds for the remainder of the proof of Theorem 10:

631 **Lemma 5.** *Let*

$$f(\boldsymbol{\theta}, W) = \frac{1}{N}\sum_{i\in[N]}\ell(\mathbf{x}_i, y_i; \boldsymbol{\theta}) + \lambda\left(-\operatorname{Tr}(WP_{\hat{y}}W^T) + 2\operatorname{Tr}(WP_{\hat{y},s}P_s^{-1/2}) - 1\right)$$

$$:= \frac{1}{N}\sum_{i\in[N]}g(\boldsymbol{\theta}, W, \mathbf{x}_i, y_i).$$

632 *Then*

633   *1. $f$ is $\beta$-smooth, where $\beta = \beta_l + 8\lambda D^2\beta_y + 4\lambda\frac{1}{\sqrt{\hat{p}_s^{\min}}}\left(\sqrt{m}k^{3/2}D\beta_{ys}\right) + 2\lambda +$*

634       *$4\lambda\left(DL_y + \frac{L_{ys}}{\sqrt{\hat{p}_s^{\min}}}\right)$.*

635   *2. $f(\boldsymbol{\theta}, \cdot)$ is $2\lambda\hat{p}_{\hat{y}}^{\min}$-strongly concave for all $\boldsymbol{\theta}$.*

636   *3. $\|W^*\|_F \leq D$, where $D$ is as defined in Theorem 10 and $W^* \in \arg\max_{W\in\mathbb{R}^{k\times m}}$ denotes*
637       *any maximizer of $f(\boldsymbol{\theta}, W)$.*

*Proof.* By Assumption 1, $g$ is twice continuously differentiable. Hence for part 1, it suffices to upper bound the spectral norm of the second derivative of $g(\cdot, \cdot, \mathbf{z})$ by $\beta$ for all $\mathbf{z} = (\mathbf{x}, y)$, where we vectorize and then differentiate with respect to $w := \operatorname{vec} W$ and/or $\boldsymbol{\theta}$, so that the resulting first and second derivatives are always vectors or a matrices (not tensors). Notice that $g(\boldsymbol{\theta}, w, \mathbf{z}) = \ell(\mathbf{z}, \boldsymbol{\theta}) - \lambda w^T(P_{\hat{y}}\otimes\mathbf{I})w + 2\lambda(\operatorname{vec}(W))^T P_{\hat{y},s}P_s^{-1/2} - \lambda$ and

$$\nabla^2 g(\boldsymbol{\theta}, w, \mathbf{z}) = \begin{pmatrix}\nabla_{\theta\theta}^2 g(\boldsymbol{\theta}, w, \mathbf{z}) & \nabla_{\theta w}^2 g(\boldsymbol{\theta}, w, \mathbf{z}) \\ \nabla_{w\theta}^2 g(\boldsymbol{\theta}, w, \mathbf{z}) & \nabla_{ww}^2 g(\boldsymbol{\theta}, w, \mathbf{z})\end{pmatrix}.$$

Further, by the definition of operator norm, we have

$$\|\nabla^2 g(\boldsymbol{\theta}, w, \mathbf{z})\|_2 \leq \|\nabla_{\theta\theta}^2 g(\boldsymbol{\theta}, w, \mathbf{z})\|_2 + 2\|\nabla_{\theta w}^2 g(\boldsymbol{\theta}, w, \mathbf{z})\|_2 + \|\nabla_{ww}^2 g(\boldsymbol{\theta}, w, \mathbf{z})\|_2.$$

638 Now we vectorize all matrices and then compute derivatives of $g$ with respect to $\theta$ and $\operatorname{vec}(W)$:

$$\nabla_\theta g(\boldsymbol{\theta}, w, \mathbf{z}) = \nabla_\theta \ell(\mathbf{z}, \boldsymbol{\theta}) - 2\lambda \nabla_\theta \operatorname{vec}(P_{\widehat{y}})^T \operatorname{vec}(W^T W) + 2\lambda \nabla_\theta \operatorname{vec}(P_{\widehat{y},s})^T \operatorname{vec}(W^T P_s^{-1/2})$$

$$(31)$$

$$= \nabla_\theta \ell(\mathbf{z}, \boldsymbol{\theta}) - 2\lambda \left[ \sum_{l \in [m], i \in [k]} W_{i,l}^2 \nabla_\theta \left( (P_{\widehat{y}})_{l,l} \right) \right]$$

$$+ 2\lambda \left[ \sum_{j \in [m], i \in [k]} W_{i,j} (\nabla_\theta (P_{\widehat{y}s})_{j,i}) (P_s^{-1/2})_{i,i} \right]; \qquad (32)$$

$$\nabla_w g(\boldsymbol{\theta}, w, \mathbf{z}) = -2\lambda W P_{\widehat{y}} + 2\lambda P_s^{-1/2} P_{\widehat{y},s}^T. \qquad (33)$$

639    Differentiating again yields:

$$\nabla_{ww}^2 g(\boldsymbol{\theta}, w, \mathbf{z}) = -2\lambda P_{\widehat{y}} \otimes \mathbf{I}_k;$$

$$\nabla_{w\theta}^2 g(\boldsymbol{\theta}, w, \mathbf{z}) = \frac{\partial}{\partial \theta} \frac{\partial g(\boldsymbol{\theta}, w, \mathbf{z})}{\partial w} = -2\lambda (\mathbf{I}_m \otimes W) \nabla_\theta P_{\widehat{y}} + 2\lambda (\mathbf{I}_m \otimes P_s^{-1/2}) \nabla_\theta \operatorname{vec}(P_{\widehat{y},s}^T);$$

$$\nabla_{\theta\theta}^2 g(\boldsymbol{\theta}, w, \mathbf{z}) = \nabla_\theta^2 \ell(\mathbf{z}, \boldsymbol{\theta}) - 2\lambda \left[ \sum_{l \in [m], i \in [k]} W_{i,l}^2 \nabla_{\theta\theta}^2 \left( (P_{\widehat{y}})_{l,l} \right) \right]$$

$$+ 2\lambda \left[ \sum_{j \in [m], i \in [k]} W_{i,j} (\nabla_{\theta\theta}^2 (P_{\widehat{y}s})_{j,i}) (P_s^{-1/2})_{i,i} \right].$$

640    Then to establish part 1, use Assumption 1, Clairaut's theorem, the definitions of the matrices
641    and fact that their entries are in $[0, 1]$, the relations $\|AB\|_2 \leq \|A\|_2 \|B\|_2$ and $\|\operatorname{vec} W\|_1 \leq$
642    $\sqrt{mk} \|\operatorname{vec} W\|_2 = \sqrt{mk} \|W\|_F$, and the fact that $\|A \otimes B\|_2 = \|A\|_2 \|B\|_2$ to bound the spec-
643    tral norm of each second derivative above.

644    The strong concavity statement follows by noticing $\nabla_{ww}^2 g(\boldsymbol{\theta}, W) \preccurlyeq -\mu \mathbf{I}$ iff $P_{\widehat{y}} \succcurlyeq \frac{\mu}{2\lambda} \mathbf{I}$ iff
645    $\min_{i \in [m]} p_{\widehat{y}}(i) \geq \frac{\mu}{2\lambda}$.

646    Part 3 follows from the expression for $W^*$ in the proof of Theorem 4.    □

647    **Lemma 6.** *Consider $f$ and $g$ as defined above. Then we have*

$$\mathbb{E}_{\mathbf{z}} \nabla g(\boldsymbol{\theta}, W, \mathbf{z}) = \nabla f(\boldsymbol{\theta}, W),$$

$$\mathbb{E}_{\mathbf{z}} \|\nabla g(\boldsymbol{\theta}, W, \mathbf{z}) - \nabla f(\boldsymbol{\theta}, W)\|_2^2 \leq 2 \left( L_\ell + 2\lambda \widetilde{L}_y D^2 + 4\lambda \frac{D}{\sqrt{\hat{p}_s^{\min}}} \sqrt{mk} \widetilde{L}_{ys} \right)^2$$

$$+ 2 \left( 2\lambda D + 2(\hat{p}_s^{\min})^{-1/2} \sqrt{mk} \right)^2,$$

648    *where both expectations are with respect to the empirical distribution on $\{\mathbf{z}_i\}_{i \in [N]}$.*

649 *Proof.* The first statement is obvious. The second follows from Eq. (32) in the proof of Lemma 5,
650 since

$$\mathbb{E}_{\mathbf{z}}\|\nabla g(\boldsymbol{\theta}, W, \mathbf{z}) - \nabla f(\boldsymbol{\theta}, W)\|_2^2$$

$$= \frac{1}{N}\sum_{i=1}^{N}\|\nabla g(\boldsymbol{\theta}, W, z_i)\|_2^2 - \frac{1}{N^2}\sum_{i,j=1}^{N}\langle\nabla g(\boldsymbol{\theta}, W, \mathbf{z}_i), \nabla g(\boldsymbol{\theta}, W, \mathbf{z}_j)\rangle$$

$$\leq 2\sup_{\mathbf{z}_i}\|\nabla g(\boldsymbol{\theta}, W, \mathbf{z}_i)\|_2^2$$

$$\leq 2\sup_{\mathbf{z}}\left\{\|\nabla_\theta g(\boldsymbol{\theta}, W, \mathbf{z})\|^2 + \|\nabla_w g(\boldsymbol{\theta}, W, \mathbf{z})\|^2\right\}$$

$$\leq 2\sup_{\mathbf{z}}\left\{\left\|\nabla_\theta \ell(\mathbf{z}, \boldsymbol{\theta}) - 2\lambda\left[\sum_{l\in[m],i\in[k]} W_{i,l}^2 \nabla_\theta\left((P_{\widehat{y}})_{l,l}\right)\right]\right.\right.$$

$$\left. + 2\lambda\left[\sum_{j\in[m],i\in[k]} W_{i,j}(\nabla_\theta\left(P_{\widehat{y}s}\right)_{j,i})\left(P_s^{-1/2}\right)_{i,i}\right]\left.\right\|_2^2\right\}$$

$$+ 2\left\|-2\lambda W P_{\widehat{y}} + 2\lambda P_s^{-1/2} P_{\widehat{y},s}^T\right\|_2^2.$$

651 Then use Assumption 1 and basic norm inequalities to bound the norm of each term. □

## E Experiment details & additional results

### E.1 Model description

For all the experiments, the model's output is of the form $O = \text{softmax}(Wx + b)$. The model outputs are treated as conditional probabilities $\mathbf{p}(\widehat{y} = i|x) = O_i$ which are then used to estimate the ERMI regularizer. We encode the true class label $Y$ and sensitive attribute $S$ using one-hot encoding. We define $\ell(\cdot)$ as the cross-entropy measure between the one-hot encoded class label $Y$ and the predicted output vector $O$.

We use logistic regression as the base classification model for all experiments in Fig. 1. The choice of logistic regression is due to the fact that all of the existing approaches demonstrated in Fig. 1, use the same classification model. The model parameters are estimated using the algorithm described in Algorithm 1. The trade-off curves for FERMI are generated by sweeping across different values for $\lambda \in [0, 10000]$. The learning rates $\eta_\theta, \eta_w$ is constant during the optimization process and is chosen from the interval $[0.0005, 0.01]$ for all datasets. Moreover, the number of iterations $T$ for experiments in Fig. 1 is fixed to 2000. Since the training and test data for the Adult dataset are separated and fixed, we do not consider confidence intervals for the test accuracy. We generate ten distinct train/test sets for each one of the German and COMPAS datasets by randomly sampling $80\%$ of data points as the training data and the rest $20\%$ as the test data. For a given method in Fig. 1, the corresponding curve is generated by taking the average test accuracy on 10 training/test datasets. Furthermore, the confidence intervals are estimated based on the test accuracy's standard deviation on these 10 datasets.

To perform the experiments in Sec. 5.2 we use a a linear model with softmax activation. The model parameters are estimated using the algorithm described in Sec. 5. The data set is cleaned and processed as described in (Kearns et al., 2018). The trade-off curves for FERMI are generated by sweeping across different values for $\lambda$ in $[0, 100]$ interval, learning rate $\eta$ in $[0.0005, 0.01]$, and number of iterations $T$ in $[50, 200]$. The data set is cleaned and processed as described in (Kearns et al., 2018).

For the experiments in Sec. 5.3, we create the synthetic color MNIST as described by Li & Vasconcelos (2019). We set the value $\sigma = 0$. In Fig. 3, we compare the performance of stochastic solver (Algorithm 1) against the baselines. We use a mini-batch of size 512 when using the stochastic solver. The color MNIST data has 60000 training samples, so using the stochastic solver gives a speedup of around 100x for each iteration, and an overall speedup of around 40x. We present our results on two neural network architectures; namely, LeNet-5 (Lecun et al., 1998) and a Multi-layer perceptron (MLP). We set the MLP with two hidden layers (with 300 and 100 nodes) and an output layer with ten nodes. A ReLU activation follows each hidden layer, and a softmax activation follows the output layer.

Some general advice for tuning $\lambda$: Larger value for $\lambda$ generally translates to better fairness, but one must be careful to not use a very large value for $\lambda$ as it could lead to poor generalization performance of the model. The optimal values for $\lambda$, $\eta$, and $T$ largely depend on the data and intended application. We recommend starting with $\lambda \approx 10$. In Appendix E.4, we can observe the effect of changing $\lambda$ on the model accuracy and fairness for the COMPAS dataset.

### E.2 More comparison to (Mary et al., 2019)

The algorithm proposed by Mary et al. (2019) backpropagates the batch estimate of ERMI, which is biased especially for small minibatches. Our work uses a correct and unbiased implementation of a stochastic ERMI estimator; Furthermore, they do not establish any convergence guarantees, and in fact their algorithm does not converge. See Fig. 4 for the evolution of *training loss* and *test accuracy* on setup of Table 1 in (Mary et al., 2019).

### E.3 Performance in the presence of outliers & class-imbalance

We also performed an additional experiment on Adult (setup of Fig 1) with a random 10% of sensitive attributes in *training* forced to 0. FERMI offers the most favorable tradeoffs on *clean test* data, however, all methods reach a higher plateau (see Fig 5). The interplay between fairness, robustness, and generalization is an important future direction. With respect to imbalanced sensitive groups, the

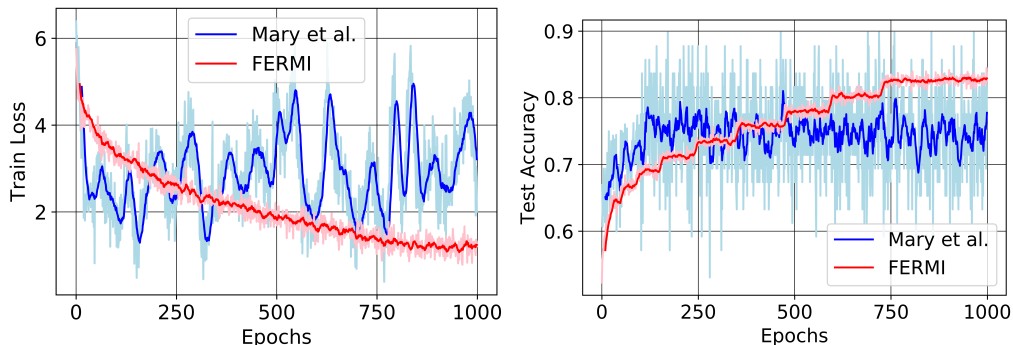

Figure 4: (Mary et al., 2019) fails to converge to a stationary point whereas our stochastic estimator easily converges

experiments in Fig 2 are on a naturally imbalanced dataset, where $\max_{s \in \mathcal{S}} p(s)/\min_{s \in \mathcal{S}} p(s) > 100$ for 3-18 sensitive attrib, and FERMI offers the favorable tradeoffs.

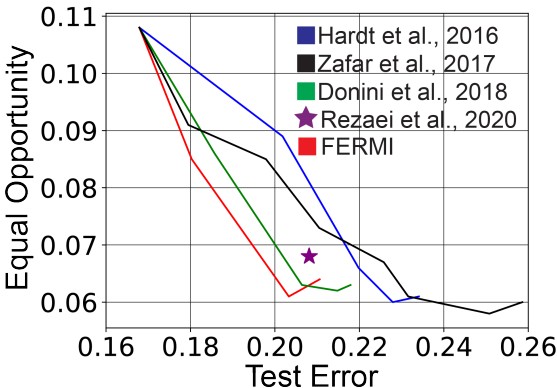

Figure 5: Comparing FERMI with other methods in the presence of outliers (random 10% of sensitive attributes in *training* forced to 0. FERMI achieves better trade-off.

### E.4 Effect of hyperparameter $\lambda$ on the accuracy-fairness tradeoffs

We run ERMI algorithm for the binary case to COMPAS dataset to investigate the effect of hyperparameter tuning on the accuracy-fairness trade-off of the algorithm. As it can be observed in Fig. 6, by increasing $\lambda$ from 0 to 1000, test error (left axis, red curves) is slightly increased. On the other hand, the fairness violation (right axis, green curves) is decreased as we increase $\lambda$ to 1000. Moreover, for both notions of fairness (demographic parity with the solid curves and equality of opportunity with the dashed curves) the trade-off between test error and fairness follows the similar pattern. To measure the fairness violation, we use demographic parity violation and equality of opportunity violation defined in Section equation 5 for the solid and dashed curves respectively.

### E.5 Complete version of Figure 1 (with pre-processing and post-processing baselines)

In Figure 1 we compared FERMI with several state-of-the-art in-processing approaches. In the next three following figures we compare the in-processing approaches depicted in Figure 1 with pre-processing and post-processing methods including (Hardt et al., 2016; Kamiran et al., 2010; Feldman et al., 2015).

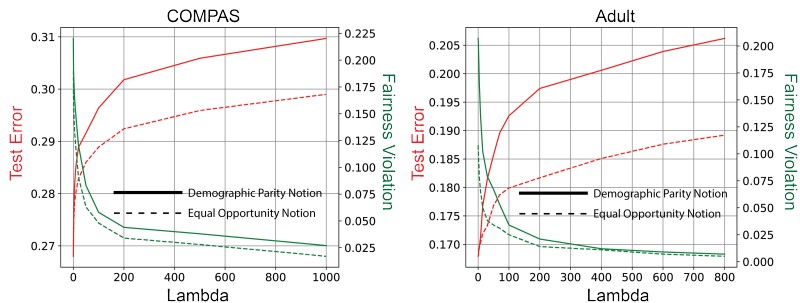

Figure 6: Tradeoff of fairness violation vs test error for FERMI algorithm on COMPAS and Adult datasets. The solid and dashed curves correspond to FERMI algorithm under the demographic parity and equality of opportunity notions accordingly. The left axis demonstrates the effect of changing $\lambda$ on the test error (red curves), while the right axis shows how the fairness of the model (measured by equality of opportunity or demographic parity violations) depends on changing $\lambda$.

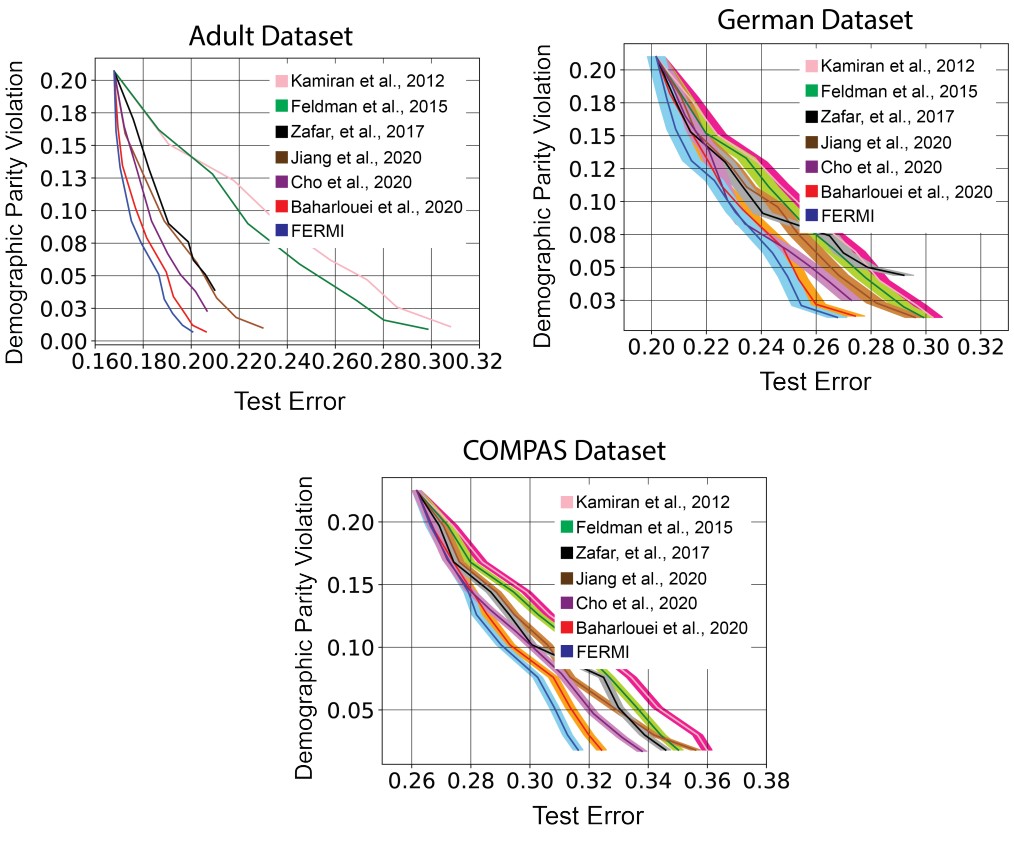

Figure 7: Tradeoff of demographic parity violation vs test error for FERMI algorithm on COMPAS, German, and Adult datasets.

### E.6 Description of datasets

All of the following datasets are publicly available at UCI repository.

**German Credit Dataset.**[3] German Credit dataset consists of 20 features (13 categorical and 7 numerical) regarding to social, and economic status of 1000 customers. The assigned task is to classify customers as good or bad credit risks. Without imposing fairness, the DP violation of the

---

[3]`https://archive.ics.uci.edu/ml/datasets/statlog+(german+credit+data)`

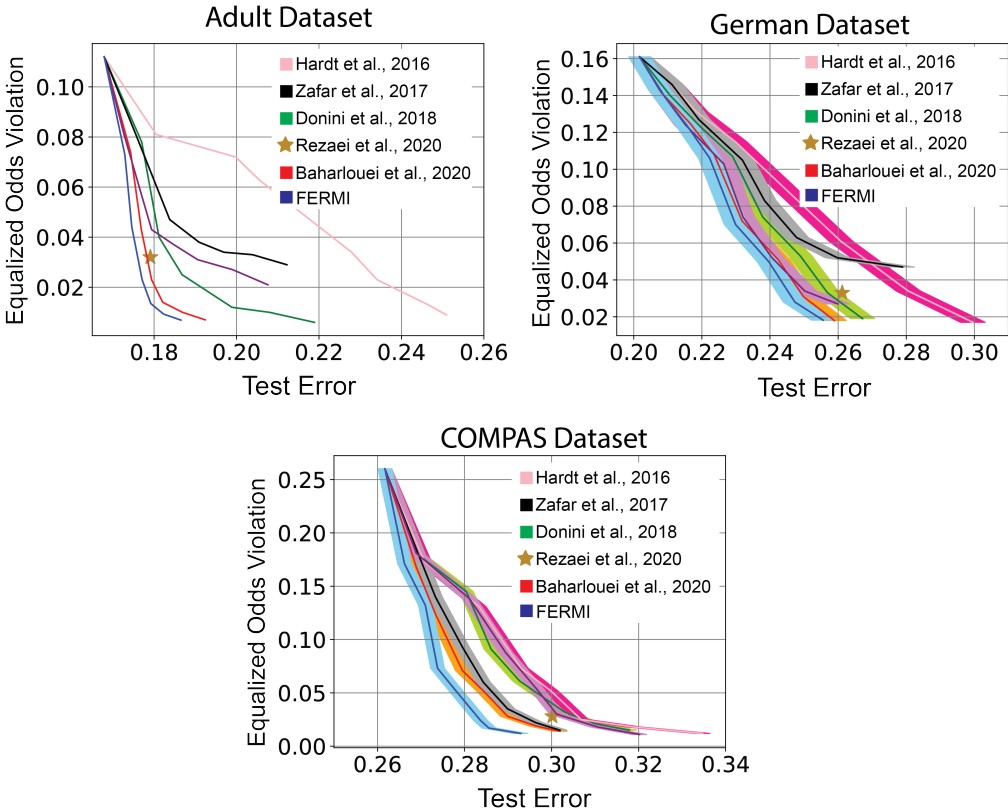

Figure 8: Tradeoff of equalized odds violation vs test error for FERMI algorithm on COMPAS, German, and Adult datasets.

trained model is larger than $20\%$. We choose $80\%$ of customers as the train data and the remaining $20\%$ customers as the test data. The sensitive attributes are gender, and marital-status.

**Adult Dataset.**[4] Adult dataset contains the census information of individuals including education, gender, and capital gain. The assigned classification task is to predict whether a person earns over 50k annually. The train and test sets are two separated files consisting of $32,000$ and $16,000$ samples respectively. We consider gender and race as the sensitive attributes (For the experiments involving one sensitive attribute, we have chosen gender). Learning a logistic regression model on the training dataset (without imposing fairness) shows that only 3 features out of 14 have larger weights than the gender attribute. Note that removing the sensitive attribute (gender), and retraining the model does not eliminate the bias of the classifier. the optimal logistic regression classifier in this case is still highly biased. For the clustering task, we have chosen 5 continuous features (Capital-gain, age, fnlwgt, capital-loss, hours-per-week), and $10,000$ samples to cluster. The sensitive attribute of each individual is gender.

**Communities and Crime Dataset**.[5] The dataset is cleaned and processed as described in (Kearns et al., 2018). Briefly, each record in this dataset summarizes aggregate socioeconomic information about both the citizens and police force in a particular U.S. community, and the problem is to predict whether the community has a high rate of violent crime.

**COMPAS Dataset**.[6] Correctional Offender Management Profiling for Alternative Sanctions (COMPAS) is a famous algorithm which is widely used by judges for the estimation of likelihood of reoffending crimes. It is observed that the algorithm is highly biased against the black defendants.

---

[4]`https://archive.ics.uci.edu/ml/datasets/adult.`

[5]`http://archive.ics.uci.edu/ml/datasets/communities+and+crime`

[6]`https://www.kaggle.com/danofer/compass`

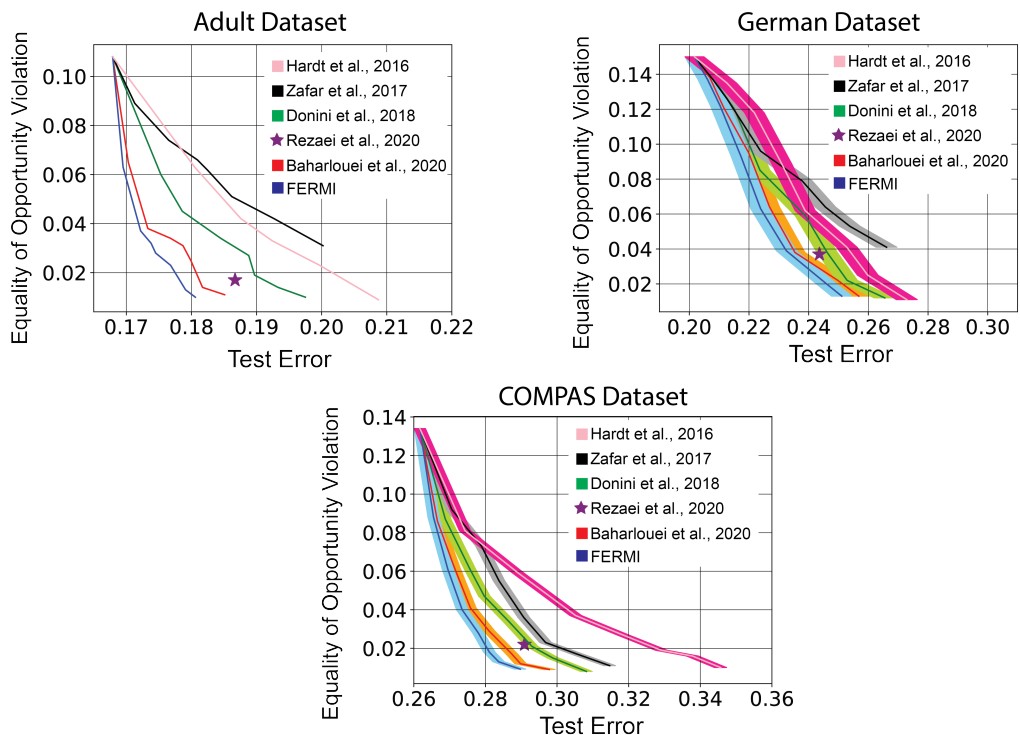

Figure 9: Tradeoff of equality of opportunity violation vs test error for FERMI algorithm on COMPAS, German, and Adult datasets.

The dataset contains features used by COMPAS algorithm alongside with the assigned score by the algorithm within two years of the decision.

**Colored MNIST Dataset.**[7] We use the code by Li & Vasconcelos (2019) to create a Colored MNIST dataset with $\sigma = 0$. We use the provided LeNet-5 model trained on the colored dataset for all baseline models of Baharlouei et al. (2020); Mary et al. (2019); Cho et al. (2020b) and FERMI, where we further apply the corresponding regularizer in the training process.

---

[7] https://github.com/JerryYLi/Dataset-REPAIR/

## F  Anonymized code for experiments

The anonymized code for all of the experiments in this paper is available on Dropbox: https://www.dropbox.com/sh/516cm8olq0idpsd/AADD0LOcPWpx4AAhzsEkFTOca?dl=0