# OpenReview forum: "FERMI: Fair Empirical Risk Minimization via Exponential Rényi Mutual Information"
_NeurIPS.cc/2021/Conference — NeurIPS 2021 Submitted_

### Official Review · Reviewer_4DYT · 2021-07-13

**Rating:** 6
**Confidence:** 4

**Summary:**

The paper explores fair empirical risk minimization through ERMI regularization. In particular, the paper provides an approach a stochastic algorithm to minimize the regularized objective with convergence guarantees.

Main Contributions:
 - Thm. 1-3: Shows that ERMI fairness violation upper bounds Shannon information, Pearson correlation, and $ L_{\infty} $ fairness violation definitions.
 - Thm. 4: Gives an unbiased estimator of ERMI regularization term.
 - Alg 1. + Thm 5.: Method of minimization which converges.
 - A number of experimental test verifying the theoretical results.

**Ethical Concerns:**

No ethical concerns.

**Limitations And Societal Impact:**

Limitation and societal impact adequately addressed.

**Main Review:**

Overall, the provides a good exploration of fair empirical risk minimization through ERMI regulation. The clarity and structure of the paper is very good, with the progression through theoretical results and then the experiments following logically.

Originality & Significance: The work seems to be significant, with Algo. 1 providing convergence guarantees on for fair empirical risk minimization. Furthermore, the results are experimentally compared to a large number of baselines and performs better despite Thm. 1-3 only giving (possibly loose) upper bounds.

Quality & Clarity: The structure and clarity of the paper is very good. Both the theoretical and experimental results seem to be sound.

Strengths:
 - Thm 1-3 motivate (FRMI obj.) nicely.
 - Alg 1. and its guarantee seems significant.
 - The experimental verification seems extensive.

Weaknesses/Suggestions/Questions:
 - Line 113: "proudct" -> "product"
 - Experiments (including SI) don't seem to include details of the machine used to run the experiments.
 - Furthermore, a rough runtime comparison between baselines and Alg 1. would be useful to add.
 - I think an informal disclaimer for the differentiability, smoothness, and Lipschitz assumption on $ \ell $ should be added to Thm 5.

Edit:
After quite a bit of reviewer discussion, I have come to the conclusion of lowering my score regarding the paper. The following are my main reasons:

  - I believe that I have overestimated the novelty of Thm 1-3 after some discussions with other reviewers. In particular, Thm 1 is only a minor change from well known results (Sason and Verdu, Eq. (94)) and Thm 3 is Pinsker's inequality with Thm 1. Although having these results being completely novel is not necessary for this paper, these results need additional clarification and reframing with respect to their novelty.

  - I missed the missing assumption / known distribution needed for the unbiased estimator, which is a major technical limitation of the work. I would argue that this assumption is not negligible. Although the authors have mentioned in other rebuttals that $ P_{s} $ "[...] in a certain population is usually public knowledge", I find that this is a rather troubling normative assumption to hold given the nature of sensitive attributes. I do believe that there is an improvement from assuming the joint distribution. However, with this, a detailed discussion of the implication of the assumption (+ its improvement from prior work) is warranted.

 - In addition to the clarification of assumption and the additional comments picked up by other reviewers (e.g., precision of large-scale models), I believe there is a non-trivial amount of restructuring which would be required to clarify the above points regarding related work and contributions.

Overall, I would like to note that I am still positive (high 6) about the paper with its unbiased estimator and strong experimental result despite the flaws listed above.


**Time Spent Reviewing:**

7

---

> ### Author Response · Authors · 2021-08-10
> **Thank you for observing the strength of our paper. We will address your comments in our revision.**
>
> Thank you for observing our theoretical and empirical contributions. Below is our response to your comments.
>
>
> _Line 113: "proudct" -> "product"_
>
> Thank you for catching this, we will fix it.
>
> =========
>
>
> _Experiments (including SI) don't seem to include details of the machine used to run the experiments._
>
> We will add this.
>
> =========
>
> _Furthermore, a rough runtime comparison between baselines and Alg 1. would be useful to add._
>
> Great suggestion! We will add this to our experiments.
>
> =========
>
> _I think an informal disclaimer for the differentiability, smoothness, and Lipschitz assumption on ℓ should be added to Thm 5._
>
> Thank you for pointing this out. We will add this.

---

> > ### Comment · Reviewer_4DYT · 2021-09-01
> > **Update in score**
> >
> > I would like to flag the authors that I have updated my score after reviewer discussion. I would like to note that although I am still positive about the paper, there have been a couple of points in discussion and reviews which has caused me to lower my score (see edits in review above).

---

### Official Review · Reviewer_GUFw · 2021-07-15

**Rating:** 6
**Confidence:** 4

**Summary:**

The authors examine the demographic parity, equalized odds, and equality of opportunity fairness constraints, and show that the Exponential Rényi Mutual Information of various random variables related to these quantities is a unifying continuous relaxation of these concepts.  Moreover they show that the FERMI upper bounds Shannon Mutual information, Rényi correlation, and Lq fairness violations. Finally, they claim an unbiased estimator for the FERMI, which seems to be the main advantage of FERMI over (non-exponential) Rényi Mutual information, and with it derive a stochastic gradient descent based algorithm for fair machine learning. Finally they close with experiments showing that FERMI controls fairness with similar effectiveness to prior results.

**Limitations And Societal Impact:**

I express concerns as to the looseness of the relationship between fairness violation concepts, and the resulting impact on model quality (and difficulty of satisfying fairness constraints) in the main review.

**Main Review:**

This is not the first paper to use FERMI in a fairness context, however here we see more technical rigor, and there are certainly new ideas.
I take issue with a technical points regarding unbiasedness that I would like to see addressed.
I also find the sample complexity discussion quite limited and unsatisfying.
I also have concerns over whether ERMI is chosen for its computational convenience, or its inherent value as a fairness concept. This is not an issue of correctness, but rather a criticism of how significant or relevant this concept really is.
Overall, the paper is good, but unless my concerns can be addressed, I do not think it meets the high bar of a NeurIPS publication.

Unbiasedness:

The discussion around theorem 4 and lemma one I find unsatisfying.  The $P_s$ matrix of course can be estimated unbiasedly from a sample, but this does not necessarily lead to an unbiased estimator.
This is rather like claiming one can derive an unbiased estimator for the variance of sums of dependent random variables. Given the expectation, this is rather straightforward, but without it, even though an unbiased expectation estimator is easily derived, it does not translate into an unbiased variance estimator!
At best, it yields an unbiased estimator for the ERMI with respect to the empirical measure of the sample, rather than the underlying distribution over samples.
This is important for the correctness of their stochastic gradient descent model, but does not have the same importance to the analysis of the overfitting of the model, specifically w.r.t. overfitting to fairness.

Furthermore, no proof of lemma 1 is given, and I am skeptical about any kind of unbiased estimator using a max over matrices $W$ (this seems to be a convex nonlinear operator).  This is not impossible (e.g., the Rademacher average or weak variance), but it is not inevitable (e.g. the wimpy variance) for estimators involving suprema to be unbiased.  I would like to see further substantiation of this claim.

I don't have a problem claiming that we have a convergent or consistent estimator for ERMI, but I want to see all sources of bias and estimation error fully analyzed.



Sample complexity:

I disagree with the statement on 311: “Furthermore, the sample complexity of estimating Rényi mutual information of order 2 (and consequently that of ERMI) scales as Θ( |S|).”  Composition of unbounded functions with the exponential operator does not preserve any definition of sample complexity of which I am aware, and if the range of RMI is |S|, I would conjecture an increase to roughly Θ( |S| exp(|S|) ) to estimate ERMI (or would appreciate an argument as to why this is not the case).  Furthermore, this is all for a single function, and presumably uniform convergence and related techniques would be required to bound ERMI over an entire family.


Choice of ERMI:

I find the specific use of ERMI a bit unsatisfying and unmotivated. My inference is that it was selected as it leads to a simple unbiased estimator, which was convenient for the SGD algorithm, rather than for its fundamental fairness properties. The claim “We show that ERMI is a stronger notion of fairness violation than... existing notions” is valid, but not necessarily a positive; one could equally well claim it is a loose bound on existing fairness concepts.
While the authors do relate ERMI to various quantities of interest, the relations are not as sharp as one may desire.

In particular:
From a (pseudo)dimensional analysis (technically it’s all unitless) perspective, I do not find theorems 1, 2, or 3 particularly surprising or interesting. The exponential in the exponential Rényi information Mutual information is very powerful, and it is unsurprising that it allows one to bound the remaining (essentially logarithmic) information quantities.
I suppose the results might be interesting when we consider a small range of possible $\hat{Y}$ and $S$, in which case the exponential can't do too much, but in general I would be much more interested in seeing what can be done with (non-exponential) Rényi mutual informations. Theorem one essentially does this, but similar hold for any RMI, theorem 2, I have little insight on but I would be interested in further exploring the idea behind the $|S| = 2$ special case, and theorem 3 in particular seems like there should be some connection between the $q$ and $\alpha$ parameters, or at least a family of similar results for arbitrary $\alpha$, where the square root is a consequence of the $\alpha=2$ choice in ERMI.


Specifics:



In table 1, you make a point that controlling EOD is stronger than EOP, which may harm accuracy.  I agree with this point, however can it not be applied to FERMI in general, as compared to the “less strong” fairness violation metrics you discuss?

Line 39 is a bit of a non-sequitur: measuring degree is important in and of itself, and these estimates are subject to statistical error as well. For hard properties, null hypothesis testing is perfectly valid way of relating samples to distributions, just as tail bounds and sample complexity analyze the statistics of soft properties.
In fact, I would argue that null hypothesis testing arose to answer questions like whether a particular binary property holds on a distribution given a sample. Developing continuous properties analogues of such discrete properties is of orthogonal interest.

71: This is not necessarily a good thing; again EOD, is “stronger than” EOP.

76 Convergence in samples or convergence in computation? I believe the latter but the writing is unclear.

82: Does large-scale mean large training set? If so what does that have to do with minibatch size?  If you mean the size of individual instances, perhaps, but in the practical world, are we not generally memory-constrained by the size of the model?

Line 115 “for brevity” sentence is unclear.



Thm. 1: Stronger is one way to put it, but one could also refer to it as looser. Is there an argument that one should be more interested in the Rényi information itself rather than the Shannon information?


Line 164 remark: I disagree with this conclusion. While it is true that bounding ERMI is sufficient when possible, it may not always be possible to control the ERMI without huge sacrifices to accuracy. I would either need to be convinced that the gap between ERMI and other fairness concepts is small, or that in situations where other fairness concepts are small but ERMI  is large, the model is, in some sense not captured by alternative fairness concepts, deeply unfair.


171: Why call it FRMI rather than FERMI?


*Score raised from 4 to 6 based on author response.*


**Time Spent Reviewing:**

13

---

> ### Author Response · Authors · 2021-08-10
> **We will clarify the points of confusion**
>
> Thank you for your detailed and constructive feedback. Below is our response to your comments.
>
> _Summary: […] Finally they close with experiments showing that FERMI controls fairness with similar effectiveness to prior results._
>
> Your summary is mostly accurate, but we take issue with the last sentence. Namely, our experiments (Section 5) do not merely show “similar” effectiveness of FERMI compared with prior results, but that FERMI achieves clear and consistent superior fairness-accuracy tradeoffs compared with SOTA across a variety of experimental settings.
>
> =========
>
> _The discussion around theorem 4 and lemma one I find unsatisfying. The Ps matrix of course can be estimated unbiasedly from a sample, but this does not necessarily lead to an unbiased estimator [...]_
>
> **We assume that $P_s$ is known** (as we use $P_s$ in Lemma 1, see lines 200-202). We will highlight this by adding it as an explicit assumption. This assumption is not unrealistic for many fairness contexts: for instance, if the sensitive attribute is race/gender, then the percentage of people of different races/genders in a certain population is usually public knowledge. So, even if the number of samples in our dataset is small and $P_s$ cannot be accurately estimated from the dataset, it is often known anyway.  Furthermore, this is much milder than estimating the joint distribution of $\hat{Y}$ and $S$ from the data, which is essentially what the baselines, such as Baharlouei et al 2020, Mary et al. 2019, and Cho et al. 2020 do, and empirically has resulted in SOTA results even with relatively large number of sensitive attributes.
>
> =========
>
> _Furthermore, no proof of lemma 1 is given, and I am skeptical_
>
>
> You are correct, there was a mis-presentation of the result. What we meant to write for Lemma 1 was that $\psi_i(\theta,w)$ (if drawn randomly from the joint distribution of data, sensitive attribute, and predicted value) in line 204 is an unbiased estimator of the quantity $ - \text{Tr}(W P_{\hat{y}} W^T) + 2 \text{Tr}(W P_{\hat{y}, s} P_{s}^{-1/2}) - 1$ in the RHS of (4). The proof of this “revised Lemma 1” follows straightforwardly from the expression of the RHS of (4). Note that by "revised Lemma 1," we have an unbiased estimator of the gradients of ERMI, which is enough to ensure that our main result, Theorem 5 (and its proof) still hold.
>
> =========
>
>
> _I disagree with the statement on 311 [...] this is all for a single function, and presumably uniform convergence and related techniques would be required to bound ERMI over an entire family._
>
> You are correct that in general the sample complexity of the two problems could be very different. However, we are interested in the regime that ERMI is small, and hence MI of order 2 ($I_2$) is small. We are also interested in the convergence of the sample complexity where estimation error is small. In particular, if $I_2 \leq 1$, and if $\Delta I_2 \leq 1$, then
>
> $$\Delta\text{ERMI} =  \exp(I_2 +\Delta I_2) - 1 - (\exp(I_2) - 1) \leq (1+ 2 \Delta I_2) \exp(I_2)- \exp(I_2) \leq 2e \Delta I_2.$$
>
> Thus in this regime, we obtain the same sample complexity as MI of order 2.
>
> =========
>
> _I find the specific use of ERMI a bit unsatisfying and unmotivated. My inference is that it was selected as it leads to a simple unbiased estimator, which was convenient for the SGD algorithm, rather than for its fundamental fairness properties._
>
> You are partially correct that ERMI was selected as a regularizer partly because it leads to an unbiased gradient estimator that allows for stochastic optimization. We will clarify the motivation behind ERMI--emphasizing the fact that it allows for stochastic optimization--in the final version. Another reason we use ERMI is that it upper bounds existing notions of fairness violation (see Thms 1-3, 8-9), which ensures that making ERMI small is enough to ensure small fairness violation with respect to existing notions. **This helps explain why FERMI empirically outperforms SOTA even when fairness violation is measured in terms of notions other than ERMI, and even when all baselines are used in batch GD form, e.g., Fig 1 in Section 5.1.**
>
> =========
>
> _The claim “We show that ERMI is a stronger notion of fairness violation than... existing notions” is valid, but not necessarily a positive [...]_
>
>  An upper bound (even a loose one) on other fairness violation notions is enough to ensure that any algorithm which makes ERMI small will be “fair” with respect to these other notions. We will change the language “is stronger than” to “upper bounds.”
> The slack in the bounds is natural in our context and indeed not the point (See Remark in lines 164-8 for discussion of the tightness of the bounds). Notice that there is no universally accepted notion of fairness violation in the literature. The existing notions of fairness violation are not necessarily comparable (e.g., mutual information and conditional demographic parity $L_\infty$ violation). This raises the question of which one (if any) to select in practice for imposing fairness. A natural approach is to use a violation notion that subsumes (bounds from above) all of these existing notions of fairness violation. Using a regularizer that bounds all notions leads to a trained model that is almost universally accepted under different notions of fairness proposed in the community.
> The fact that this **“stronger/upper bound” notion of fairness violation does not hurt accuracy** is evidenced empirically by our superior fairness-accuracy tradeoffs vs. SOTA across an extensive range of experiments.
>
> =========
>
>
> _From a (pseudo)dimensional analysis (technically it’s all unitless) perspective, I do not find theorems 1, 2, or 3 particularly surprising or interesting. The exponential in the exponential Rényi information Mutual information is very powerful, and it is unsurprising that it allows one to bound the remaining [...]_
>
> We want to emphasize that one of our main contributions is the unbiased estimator of ERMI, which allows for the first provably convergent stochastic fairness algorithm. Without the exponential, it is unclear that we would be able to derive such an estimator. Also, your comment about Theorems 1-3 being due to the exponential is partially correct, but not entirely. For example, log of ERMI also bounds mutual information (MI), but exponential Rényi correlation does not bound MI.
>
> =========
>
> _In table 1, you make a point that controlling EOD is stronger than EOP, which may harm accuracy. I agree with this point, however can it not be applied to FERMI in general, as compared to the “less strong” fairness violation metrics you discuss?_
>
> In theory, it is indeed not clear in general that using an upper-bound notion of fairness violation (e.g. ERMI) as a regularizer would lead to a more accurate classifier. **However, our experiments demonstrate that FERMI consistently achieves superior accuracy-tradeoff curves vs. SOTA in-processing methods which employ weaker regularizers.**
>
> =========
>
> _Line 39 is a bit of a non-sequitur [...]_
>
> You make a valid point. We will revise.
>
> =========
>
> _ 71: This is not necessarily a good thing [...]_
>
>  Please see our previous response about this issue.
>
> =========
>
> _76 Convergence in samples or convergence in computation? I believe the latter but the writing is unclear._
>
> Convergence in computation. We will clarify the wording.
>
> =========
>
> _82: Does large-scale mean large training set? If so what does that have to do with minibatch size? If you mean the size of individual instances, perhaps, but in the practical world, are we not generally memory-constrained by the size of the model?_
>
> Large-scale means large training set ($N$) and large model ($|\theta|$). In such cases, using a full batch first-order method would require $N$ gradient evaluations each requiring $|\theta|$ differentiations, with runtime $O(N |\theta|)$ per iteration. This is computationally prohibitive, which is why stochastic optimization methods (e.g. SGD) are widely preferred/used in practice. In addition, stochastic methods can escape spurious local optima in nonconvex optimization (see, e.g., Kleinberg et al. 2018, https://arxiv.org/abs/1802.06175).
>
> =========
>
> _Line 115 “for brevity” sentence is unclear._
>
> This sentence got cut off: we will revise to “we present all definitions and results for $(Z,\mathcal{Z})$-fairness notion”.
>
> =========
>
>
> _Thm. 1: Stronger is one way to put it, but one could also refer to it as looser. Is there an argument that one should be more interested in the Rényi information itself rather than the Shannon information?_
>
> We want to emphasize that one of our main contributions is the unbiased estimator of ERMI, which allows for the first provably convergent stochastic fairness algorithm. Also, Theorems 1-3 connects ERMI to existing notions of fairness. Our extensive empirical studies demonstrate that this regularizer outperforms SOTA in a wide range of experiments.
>
> =========
>
>
> _Line 164 remark: I disagree with this conclusion. While it is true that bounding ERMI is sufficient when possible, it may not always be possible to control the ERMI without huge sacrifices to accuracy [...]_
>
> **Our experiments show that the use of ERMI as a regularizer leads to an algorithm that is effective in achieving the most favorable fairness-accuracy tradeoff in practice**. Also, see our earlier responses about the computational advantages (stochastic algorithm) and benefits of having an upper bound (even a loose one) on other fairness violation notions.
>
> =========
>
> _171: Why call it FRMI rather than FERMI?_
>
> This is the population level objective, not the **empirical** objective that we solve.

---

> > ### Comment · Reviewer_GUFw · 2021-08-13
> > **A Few Followups**
> >
> > Thanks for the detailed response.
> >
> > *Regarding minibatch size:*
> >
> > I agree, arguments for SGD in about escaping from local minima, and other desirable properties are appealing.
> > However, my question was specifically about the phrase “when minibatch size is small (as is practically necessary for large-scale problems).”  I don’t think dataset size impacts minibatch size, unless you meant in a relative sense?
> >
> > *We want to emphasize that one of our main contributions is the unbiased estimator of ERMI, which allows for the first provably convergent stochastic fairness algorithm. Without the exponential, it is unclear that we would be able to derive such an estimator. Also, your comment about Theorems 1-3 being due to the exponential is partially correct, but not entirely. For example, log of ERMI also bounds mutual information (MI), but exponential Rényi correlation does not bound MI.*
> >
> > This sounds very appealing but it is not obvious to me, particularly as $\ln(\text{ERMI})$ can be negative.
> > Perhaps you meant $\text{MI} \leq \ln(1+\text{ERMI})$?
> > Either way, this sounds like a stronger presentation of theorem 1, as it is clear that ERMI controls MI more strongly than currently presented (and thus a looser constraint on ERMI suffices to control MI).  I would prefer a presentation that includes this as an intermediate form in the bound.

---

> > > ### Author Response · Authors · 2021-08-13
> > > **Response to follow-ups**
> > >
> > > Many thanks for providing us with another opportunity to respond to your questions/concerns.
> > >
> > > _I agree, arguments for SGD in about escaping from local minima, and other desirable properties are appealing. However, my question was specifically about the phrase “when minibatch size is small (as is practically necessary for large-scale problems).” I don’t think dataset size impacts minibatch size, unless you meant in a relative sense?_
> > >
> > > By large-scale problems, we mean large models with number of parameters, $|\theta|$, in 100s of millions or even billions, e.g.,  BART (https://arxiv.org/abs/1910.13461), ViT (https://arxiv.org/abs/2010.11929) or GPT-2 (https://openai.com/blog/gpt-2-1-5b-release/). In such cases, in training time, the available memory on a node constrains us to use batch sizes that are no bigger than $|B| = 20$ to handle the gradient computations (the memory requirement is roughly dictated by $|\theta| \times |B| \times b,$ where $b$ is the number of bits used for representing each parameter, e.g., 16). For example, we are reciting the following "**The pretrained models are fine-tuned with a mini-batch of 6 on 8 Nvidia V100 ....**" from a recent work on dialog modeling: https://arxiv.org/pdf/2005.05298.pdf.
> > >
> > > You are correct that the dataset size is not the limiting factor here. For example, we’d face the same batch size constraint even if we were using a large pre-trained model and fine-tuning it on a small dataset (say 100 samples). We will clarify “large-scale problems” to avoid confusion.
> > >
> > > =====
> > >
> > > _This sounds very appealing but it is not obvious to me, particularly as $\ln(\text{ERMI})$ can be negative. Perhaps you meant $\text{MI} \leq \ln(1+ \text{ERMI})$? Either way, this sounds like a stronger presentation of theorem 1, as it is clear that ERMI controls MI more strongly than currently presented (and thus a looser constraint on ERMI suffices to control MI). I would prefer a presentation that includes this as an intermediate form in the bound._
> > >
> > > Yes, we exactly meant $\text{MI} \leq \ln(1+ \text{ERMI})$, sorry for being informal in presenting it. This claim follows directly from the intermediate step in Thm 1, which is recalled here:
> > > $$ 0 \leq \text{MI} \leq \text{MI}_2 \leq e^{\text{MI}_2 } - 1 = \text{ERMI}$$
> > > We agree with you that this should be more clearly remarked and communicated, as it is a tighter bound than is currently claimed. Having said that, the regime of interest for fairness is where $\text{ERMI}$ is small in which case the slack in exponentiation (third inequality) is bounded as well.

---

> ### Author Response · Authors · 2021-08-21
> **Thanks for engaging with us to resolve issues.**
>
> Dear Reviewer GUFw,
>
> We are grateful that you took time to read our response and engage in a dialog with us to resolve your remaining concerns and increase your score.
>
> $\newline$
>
> Thanks!
>
> Authors

---

### Official Review · Reviewer_VsBW · 2021-07-18

**Rating:** 5
**Confidence:** 3

**Summary:**

This paper proposes a new characterization of the exponentiated (second-order) Rényi mutual information (ERMI). It then uses an empirical version of this idealized characterization as a fairness regularizer. (Many notions of fairness can be written as conditional independence statements, whose approximation can be expressed using mutual information.) The benefit of the new characterization is that the overall optimization can be written as a min-max optimization where the objective function is sum-decomposable. This allows using stochastic gradient descent-ascent to converge to a stationary point. Experimental evidence is given that shows the new approach achieves better fairness-accuracy tradeoff, even compared to prior methods using a different empirical version of ERMI as a fairness regularizer.

**Ethical Concerns:**

No ethical concerns.

**Limitations And Societal Impact:**

Societal impact has been discussed through the topic of fairness itself.

**Main Review:**

### Strengths:

+ The characterization of ERMI in a variational form (Theorem 4, maximizer of a quadratic expression) is interesting and could be useful in other situations too. _(originality)_

+ The experimental results seem quite strong, showing clearly that there is merit in this approach. _(significance)_

### Weaknesses:

- There are a couple of issues with the claims. First, Lemma 1 as stated is not correct. It claims that the proposed ERMI estimator (the empirical version of the variational characterization) is unbiased. This is stated without proof, as an immediate consequence of the characterization itself (Theorem 4), perhaps due to the linear expression within the maximizer (if $P_s$ is indeed assumed known.)  While unbiasedness holds without the max, it’s not obvious at all why it would with the max, and very likely doesn’t. Considering this is one of the repeated selling points, it requires an explanation from the authors. _(quality)_

- Second, assuming $P_s$  known or easily learnable from the data sounds reasonable, but considering how much the convergence results hinge on the gradient being unbiased, I’m not sure it can be swept under the carpet that easily. This is particularly important because it’s the _inverse_ of $P_s$ that shows up. The problem can be particularly pronounced when there are many sensitive attributes, some of them observed rarely in the data, and inaccuracy near zero can create a large bias and variance in the inverse of the estimated $P_s$. Considering the paper makes claims of unequivocal convergence to distinguish itself from alternatives, this also needs addressing. _(quality)_

- The presentation is often unnecessarily repetitive, with the effect of selling the paper’s results rather than clearly presenting its contributions. _(clarity)_

- Most of the comparison results between Renyi mutual information and other measures of independence are previously known (Theorems 1 through 3). _(originality)_

### Suggestions:

* In the proof of Theorem 4, `line 607` and `line 608` (second equation) the expression for $W^*$ has swapped matrices. In the end everything works out, so it just needs a clean-up. Also at the end of the proof, you get $D_R + 1$.

* In Lemma 1, once clarified, `line 198` needs also a $\frac{1}{N}$ factor.

* Typos: `line 113` product, `line 119` equalized.

[Edit: Thank you authors for the active discussions. Lemma 1 is now correct, although it's not a lemma but rather a corollary to Theorem 4. The flow makes sense, in that if SGDA targets the min-max expected objective, it's true that the max expected-objective is FRMI, and your samples give you access to unbiased gradients of the min-max expected objective. You need to change references to unbiasedness throughout the paper to make sure this change is properly reflected. The strong experimental results that you have show merit, but  I believe a reworking of the theoretical insight with clarity and modesty will make a better case for the paper. I will leave my evaluation unchanged.]

**Time Spent Reviewing:**

4.5

---

> ### Author Response · Authors · 2021-08-10
> **Clarifying Lemma 1 and Assumption on $P_s$**
>
> Thank you for observing the originality of the variational form and the strength of the numerical experiments. Also, thank you for your thoughtful and detailed feedback. Below is our response to your comments.
>
>
> _There are a couple of issues with the claims. First, Lemma 1 as stated is not correct. It claims that the proposed ERMI estimator (the empirical version of the variational characterization) is unbiased. This is stated without proof, as an immediate consequence of the characterization itself (Theorem 4), perhaps due to the linear expression within the maximizer (if Ps is indeed assumed known.) While unbiasedness holds without the max, it’s not obvious at all why it would with the max, and very likely doesn’t. Considering this is one of the repeated selling points, it requires an explanation from the authors. (quality)_
>
> You are correct. What we meant to write for Lemma 1 was that $\psi_i(\theta,w)$ (if drawn randomly from the joint distribution of data, sensitive attribute, and predicted value) in line 204 is an unbiased estimator of the quantity $ - \text{Tr}(W P_{\hat{y}} W^T) + 2 \text{Tr}(W P_{\hat{y}, s} P_{s}^{-1/2}) - 1$ in the RHS of (4). The proof of this “revised Lemma 1” follows straightforwardly from the expression of the RHS of (4). Note that by "revised Lemma 1," we have an unbiased estimator of the gradients of ERMI, which is enough to ensure that our main result, Theorem 5 (and its proof) still hold.
>
> =========
>
> _Second, assuming Ps known or easily learnable from the data sounds reasonable, but considering how much the convergence results hinge on the gradient being unbiased, I’m not sure it can be swept under the carpet that easily. This is particularly important because it’s the inverse of Ps that shows up. The problem can be particularly pronounced when there are many sensitive attributes, some of them observed rarely in the data, and inaccuracy near zero can create a large bias and variance in the inverse of the estimated Ps Considering the paper makes claims of unequivocal convergence to distinguish itself from alternatives, this also needs addressing. (quality)_
>
> We do not mean to hide this assumption. We will highlight this issue by adding an explicit assumption that $P_s$ is known. This assumption is not unrealistic for many fairness contexts: for instance, if the sensitive attribute is race/gender, then the percentage of people of different races/genders in a certain population is usually public knowledge. So, even if the number of samples in our dataset is small and $P_s$ cannot be accurately estimated from the dataset, it is often known anyway. Furthermore, this is much milder than estimating the joint distribution of $\hat{Y}$ and $S$ from the data, which is essentially what the baselines, such as Baharlouei et al 2020, Mary et al. 2019, and Cho et al. 2020 do, and empirically has resulted in SOTA results even with a relatively large number of sensitive attributes.
>
> =========
>
> _The presentation is often unnecessarily repetitive, with the effect of selling the paper’s results rather than clearly presenting its contributions. (clarity)_
>
> We will go through and eliminate redundancies in order to streamline and clarify the presentation. We also appreciate it if you can please give explicit pointers.
>
> =========
>
>
> _Most of the comparison results between Renyi mutual information and other measures of independence are previously known (Theorems 1 through 3). (originality)_
>
> To the best of our knowledge, Theorems 1,3 are not observed in the literature. We have made connections with existing pieces and lemmas that we were aware of (and referenced them). Having said that, since ERMI is inherently an $f$-divergence, we would not be surprised if there are other pieces of related work that we may have missed. **We would be glad to credit any works we missed if you could point us to relevant articles or results.**
>
> =========
>
>
> _In the proof of Theorem 4, line 607 and line 608 (second equation) the expression for W∗  has swapped matrices. In the end everything works out, so it just needs a clean-up. Also at the end of the proof, you get  DR+1_
>
> Good catches, we will fix those.
>
> =========
>
>
> _In Lemma 1, once clarified, line 198 needs also a 1/N factor._
>
> Thank you very much.
>
> =========
>
>
> _Typos: line 113 product, line 119 equalized._
>
> Thank you for catching these; we will fix them.

---

> ### Author Response · Authors · 2021-08-21
> **Has our response addressed your concerns?**
>
> Dear Reviewer VsBW,
>
> We would be grateful if you can confirm whether our response has addressed your concerns, and let us know if any issues remain. To recap our response, we:
>
> $\bullet$ Revised Lemma 1 and provided a statement and proof, which is what we need in Theorem 5 to prove convergence of FERMI Algorithm (see our comment with the title "Revised Lemma 1").
>
> $\bullet$ Provided discussion around why knowledge of $P_s$ is not limiting in practice, especially compared to existing baselines. We also argued that this assumption has not hurt the empirical performance of FERMI, as it outperforms all baselines across all experiments.
>
> $\newline$
>
> Thanks!
>
> Authors

---

### Official Review · Reviewer_JwMW · 2021-07-19

**Rating:** 4
**Confidence:** 4

**Summary:**

This paper looks to extends existing work on fairness-arawe learning with continuous attributes/labels by improving estimation of (ideal) regularizer. It proposes an unbiased estimator of a regularizer based on $\\chi^2$ divergence using a variational formulation. It then proposes a set of experiments to illustrate the performance of the method.

**Ethical Concerns:**

Not that I can think of.

**Limitations And Societal Impact:**

See main review

**Main Review:**

**Core idea**
* Overall, I like the core idea of using a variational approach to obtain an unbiased estimator of the $\\chi^2$ divergence used in previous works (previous estimators not being unbiased).
* I think the paper could focus more on this aspect, in particular, showing how its estimator converges towards the $\\chi^2$ divergence, versus the previously proposed ones (ex: Mary 2019). And this, independently of its use as a fairness regulariser.

**Exposition and positioning**
* I found the paper tends to oversell its results and would generally be improved by a better and more fair discussion of its limitation and comparaison to SOTA. Few examples of imprecise/misleading things:
  * I don't see the need to define a new notion (ERMI), to then explain it is already existing: the $\\chi^2$ divergence between the joint distribution and the kronecker product of marginals.
  * The *non-binary* label/attribute positioning should be more precise, as it means "discrete" rather than "continuous". Indeed, as main SOTA compared in the experiments are designed for continuous label/attributes (Mary 2019, Cho 2020...), this imprecise naming is *misleading*. I was believing the paper treats about the continuous case until Th. 4 (end of page 5).
  * This is even more confusing when the comment after Lemma 1 compares the resulting estimator (for discrete variables) to the bias introduced by *continuous* density estimation methods introduced in (Mary 2019, Baharlouei 2020).
  * Many methods designed for binary variables have strait-forward extensions to discrete variables.
* Thm 1,2,3 show that the $\\chi^2$ divergence upper-bounds some other existing coefficients that characterize independence.
  * Some are coming from the literature, or are strait-forward.
  * I find the statements "ERMI is stronger than MI/HGR" imprecise. It is unclear what "stronger" means. It is an upper-bound, yes. But MI or HGR also characterize completely independence. So I don't see in what sense it is "stronger".
  * I'm not sure these 3 results bring anything to the paper, as the main contribution is not the use of the $\\chi^2$ divergence but rather the variational form to build an unbiased estimator.
* In Table 1: even "while satisfying eod guarantees satisfying eop [...] we only credit those works that provide/implement algorithms for a given fairness notion" -> That mis-characterizes previous work.

**Experimentation**
* Experiments seems OK overall.

**Time Spent Reviewing:**

6

---

> ### Author Response · Authors · 2021-08-10
> **Our theory informs the stochastic scalable FERMI algorithm outperforming SOTA in extensive empirical experiments**
>
> Thank you for your constructive feedback. **We want to emphasize that the goal of this paper is to devise a fairness regularizer suitable for stochastic optimization.  Equally important as the theoretical results, one of the main contributions of this paper is an extensive set of experiments across a variety of binary and non-binary problem settings.** Below is our response to your comments.
>
>
> _Overall, I like the core idea of using a variational approach to obtain an unbiased estimator of the $\chi^2$ divergence used in previous works (previous estimators not being unbiased). I think the paper could focus more on this aspect, in particular, showing how its estimator converges towards the $\chi^2$ divergence, versus the previously proposed ones (ex: Mary 2019). And this, independently of its use as a fairness regulariser._
>
> We would like to emphasize that the goal of this paper is to devise a fairness regularizer suitable for stochastic optimization. Our theoretical developments inform a stochastic fairness algorithm, FERMI. An equally important contribution of this work is the extensive empirical comparison of FERMI with SOTA algorithms. We agree with the reviewer that the techniques developed in this paper can have applications beyond fairness, which can be a research topic for future investigations.
>
> =======
>
> _I found the paper tends to oversell its results and would generally be improved by a better and more fair discussion of its limitation and comparaison to SOTA. Few examples of imprecise/misleading things: I don't see the need to define a new notion (ERMI), to then explain it is already existing: the χ2vdivergence between the joint distribution and the kronecker product of marginals._
>
>
> ERMI is indeed a particular instance of the $\chi^2$ divergence applied to the fairness problem, as we note right after defining ERMI. We believe it is more clear and concise to directly define the specialized version of this divergence for our setting, rather than recalling the definition of $\chi^2$ divergence and then explaining how we apply it in our problem. Also, “ERMI” is easier to write and say than “$\chi^2$ divergence between the joint distribution of $\hat{Y}, S$ and the kronecker product of marginals” and we refer to this quantity repeatedly throughout the paper.
>
> =======
>
> _The non-binary label/attribute positioning should be more precise, as it means "discrete" rather than "continuous". Indeed, as main SOTA compared in the experiments are designed for continuous label/attributes (Mary 2019, Cho 2020...), this imprecise naming is misleading. I was believing the paper treats about the continuous case until Th. 4 (end of page 5)._
>
>
> Thank you for your comment. We will clarify that we are working with discrete, not continuous, sensitive attributes, making this very explicit in the introduction.
>
> =======
>
> _This is even more confusing when the comment after Lemma 1 compares the resulting estimator (for discrete variables) to the bias introduced by continuous density estimation methods introduced in (Mary 2019, Baharlouei 2020)._
>
> The fairness estimators in Mary et al. and Baharlouei et al. are both biased (if used in stochastic optimization) even for **discrete** variables (see section 5.3 for an empirical investigation where we show that FERMI outperforms these baselines by 30+% when mini-batch size is smaller than 64).
>
> =======
>
> _Many methods designed for binary variables have strait-forward extensions to discrete variables._
>
> **Can you please explain what exactly is “imprecise/misleading” about our comparison/discussion of existing works, so that we can revise the paper?** In Table 1, we show the SOTA in-processing methods that can handle non-binary sensitive attributes and non-binary targets, and which provide code for non-binary. In sections 5.2  and 5.3, we compare against Mary et al., Baharlouei et al. and Cho et al. for non-binary targets and sensitive attributes.
>
> =======
>
> _Thm 1,2,3 show that the $\chi^2$ divergence upper-bounds some other existing coefficients that characterize independence. Some are coming from the literature, or are strait-forward._
>
> To the best of our knowledge Theorems 1 and 3 are not observed in the literature. We have made connections with existing pieces and lemmas that we were aware of (and referenced them). Having said that, since ERMI is inherently an $f$-divergence, we would not be surprised if there are other pieces of related work that we may have missed. **We would be glad to credit any works we missed if you could point us to relevant articles or results.**
>
> =======
>
> _I find the statements "ERMI is stronger than MI/HGR" imprecise. It is unclear what "stronger" means. It is an upper-bound, yes. But MI or HGR also characterize completely independence. So I don't see in what sense it is "stronger"._
>
> By “stronger”, we mean it provides an upper bound. We will change the language to “ERMI upper bounds MI/HGR”.
>
> =======
>
> _I'm not sure these 3 results bring anything to the paper, as the main contribution is not the use of the $\chi^2$ divergence but rather the variational form to build an unbiased estimator._
>
> While the unbiased estimation of ERMI is indeed one of our main contributions, we also argue that our **application of this estimator in fairness** and our empirical results are equally important. The point of Theorems 1-3 (and Theorems 8-9 in the Appendix) is to motivate the use of ERMI as a regularizer for (equalized odds, equal opportunity, and demographic parity) fairness problems. In particular, these 3 results imply that any algorithm that achieves small ERMI will have small fairness violations with respect to these other coefficients (see line 49-50).  This helps explain why FERMI empirically achieves favorable fairness-accuracy tradeoffs even when fairness violation is measured in terms of coefficients other than ERMI (see lines 161-3 and Section 5). We will make this line of thinking more explicit at the beginning of Section 3 to enhance the exposition and motivate these results.
>
> =======
>
> _In Table 1: even "while satisfying eod guarantees satisfying eop [...] we only credit those works that provide/implement algorithms for a given fairness notion" -> That mis-characterizes previous work._
>
> Thank you for pointing this out. We will remove this sentence and fix the characterization of previous work.
>
> =======
>
> _Experiments seems OK overall._
>
> **We want to emphasize that equally important as the theoretical results, one of the main contributions of this paper is an extensive set of experiments across a variety of binary and non-binary problem settings, which show that our novel algorithm FERMI clearly and consistently outperforms SOTA.**

---

> ### Author Response · Authors · 2021-08-21
> **Has our response addressed your concerns?**
>
> Dear Reviewer JwMW,
>
> We would be grateful if you can confirm whether our response has addressed your concerns, and let us know if any issues remain. To recap our response, we:
>
> $\bullet$ Clarified that the main goal of this paper is a fairness algorithm that is applicable to stochastic optimization with convergence guarantees. Similar to mutual information, ERMI (and especially our variational form) can be used in numerous other applications, and is an area for future work.
>
> $\bullet$ Emphasized that FERMI outperforms all baselines even when all baselines are solved using full batch gradients, showing the effectiveness of ERMI as a fairness violation.
>
> $\bullet$ Made clarifications on comparisons with related work.
>
> $\bullet$ Clarified that Theorems 1-3 motivate the use of ERMI as a fairness regularizer. We will also be stating a tighter variant of Theorem 1 thanks to clarifying questions from Reviewer GUFw.
>
> $\newline$
>
> Thanks!
>
> Authors

---

### Author Response · Authors · 2021-08-11
**Revised Lemma 1**

**Lemma 1.** Let $(\mathbf{X}, S, Y, \widehat{Y}(\mathbf{X}; \theta))$ be a random draw from $P_{\mathbf{X}, S, Y, \widehat{Y}}$.
Further, let
$$
    \psi(\mathbf{X},S, Y,\widehat{Y}; \theta,W) := -\text{Tr}(W \widehat{\mathbf{Y}}(\mathbf{X}; \theta) \widehat{\mathbf{Y}}^T(\mathbf{X}; \theta)  W^T) + 2 \text{Tr}(W \widehat{\mathbf{Y}}(\mathbf{X}; \theta) \mathbf{S}^T P_{s}^{-1/2}) - 1.
$$

Then, problem (FRMI obj.) can be written as
$$
\min_{\theta}\left(\text{FRMI}(\theta) = \max_{W \in \mathbb{R}^{k \times m}} \mathbb{E} \left[\ell(\mathbf{X},Y;\theta) + \lambda \psi(\mathbf{X},S, Y, \widehat{Y}; \theta,W)\right]\right).
$$

*Proof.* The proof simply follows the fact that

$$
\max_{W \in \mathbb{R}^{k \times m}}\mathbb{E} \left[ \psi(\mathbf{X},S, Y, \widehat{Y}; \theta,W)\right] = \max_{W \in \mathbb{R}^{k \times m}}\left(- \text{Tr}(W P_{\widehat{y}} W^T) + 2 \text{Tr}(W P_{\widehat{y}, s} P_{s}^{-1/2}) - 1 \right)= D_R(\widehat{Y}; S),
$$

where the last equality is due to Theorem 4.

---

### Decision · Program_Chairs · 2021-09-27

**Decision:**

Reject

**Comment:**

All the reviewers like the variational approach in order to obtain an unbiased estimator for ERMI, and (modulo concerns about the assumption that $P_s$ is known), the revised Lemma 1 does fix the issue with unbiasedness that was highlighted in some of the review. Also, there is agreement that the experimental results are impressive. So, from the method and experiments side, this could make for a very strong paper. However, very unfortunately, there are some significant weaknesses when it comes to a key assumption (which is related to a key selling point of the paper), some of the results not being original as a few reviewers had originally thought, and also some important issues with the current presentation. I stress that this paper can be a strong paper, and it was very close to being accepted, but ultimately the concerns are numerous enough to warrant the review of a new, restructured/revised version of the paper prior to the paper being accepted at NeurIPS. The remainder of this review focuses on the various issues that came out in the reviews and the ensuing discussion.

First — and this was not the most major factor in the decision on this paper — it seems that Theorems 1 and 3 are not novel (I acknowledge that the authors did cite Theorem 2 from Mary et al.). In the discussion phase, some reviewers gave pointers to where these results can be essentially found. For pointers for Theorems 1 and 3, please see the review of 4DYT. For Theorem 2, one reviewer mentioned that it "is already stated in Mary et al. (2019) and derives from Witsenhausen's characterization of Renyi correlation." They also mentioned that the part of the theorem stating that Renyi's coefficient upper bounds Pearson is imprecise, because Pearson's definition does not hold when $S$ is multi-dimensional (which is the case of interest here), and in 1 dimension, the inequality is by definition of Renyi's coefficient. Overall, the lack of novelty of Theorems 1 through 3 is not a severe weakness, but in light of what was already known, some changes to the presentation are in order.

One of the major criticisms is the assumption that $P_s$ is known; there was consensus among the reviewers that this is an important limitation. I especially see this as problematic since (as mentioned by Reviewer 4DYT) this can be a troubling normative assumption and (as Reviewer VsBW mentions) the inverse of $P_s$ is what we need to deal with, and so one needs to be very careful with estimation errors for $P_s$. Still, as the authors mention, previous important related works assumed the joint, and so the authors do operate under weaker assumptions. At a minimum, a detailed discussion regarding assumptions is in order. With so much of this work focusing on unbiasedness and convergence, the assumption of what is known (rather than estimated) deserves higher scrutiny and needs serious discussion in the main text of the paper.

The last point I discuss is a significant issue (whether or not the authors realize it), but I stress that this issue was not in isolation the reason for the paper not being accepted. The issue is best summarized by the following line from Reviewer VsBW (also essentially conveyed by Reviewer JwMW): "The presentation is often unnecessarily repetitive, with the effect of selling the paper’s results rather than clearly presenting its contributions." From my detailed read, there is significant repetitiveness and also some overselling is very apparent to me. Given that two reviewers and I independently came to this conclusion, I really do think this is an important issue. I urge the authors to solicit feedback from their peers if the authors are unable to see what parts of their paper are repetitive and to also consider reducing the overselling; this is an exercise that overall will benefit the authors by improving their scientific writing in the future and also which will allow them more space to present their contributions. One reviewer lamented that there was not much precision in what is being estimated, and there also was a lack of precision when discussing sample complexity in the paper. Having more space will allow you to properly discuss these important points.

I will now provide some guidance here about the repetition that I found and the overselling I found (again, in isolation the below is not the reason for the paper not being accepted, and also, it is not the major reason either, but it is also not a minor one).

Regarding overselling: The authors often mention that ERMI is stronger than other fairness measures (see Theorems 1 through 3). However, stronger here just means that if ERMI is small, the other measures are small. Basically, ERMI can be viewed as a looser upper bound. Saying "stronger" in many places, as the authors do, give the connotation that ERMI is better, while a more scientific presentation would mention how it's simply looser (yet, the authors can highlight the benefit of ERMI in their experiments, so all is not lost).

Regarding repetitiveness: In three different places, the authors mention that Mary et al. (2019) has an estimator that is biased, and they repeatedly mention that theirs is unbiased. Also, they mention exactly once that the estimator of Mary et al. is biased but consistent (which, of course, means asymptotically unbiased). The scholarly way to say this would be that the method of Mary et al. is asymptotically unbiased (not saying it this way is, to me, mild overselling, so this should also be included in the "overselling" paragraph above). They also mention 4 times in the paper, even before the experiments section, that their methods outperform those of Mary et al. (2019). This amount of repetition seems less than scholarly.

In conclusion, this paper definitely has strong merit, but it needs a serious rewriting/restructuring though in order to appear at venue of the caliber of NeurIPS. As one reviewer stated, "The strong experimental results that you have show merit, but I believe a reworking of the theoretical insight with clarity and modesty will make a better case for the paper".